# N4-acetylcytidine (ac4C) promotes mRNA localization to stress granules

Pavel Kudrin[1,2], Ankita Singh [iD][3], David Meierhofer [iD][4], Anna Kuśnierczyk[5] & Ulf Andersson Vang Ørom [iD][1✉]

## Abstract

**Stress granules are an integral part of the stress response that are formed from non-translating mRNAs aggregated with proteins. While much is known about stress granules, the factors that drive their mRNA localization are incompletely described. Modification of mRNA can alter the properties of the nucleobases and affect processes such as translation, splicing and localization of individual transcripts. Here, we show that the RNA modification N4-acetylcytidine (ac4C) on mRNA associates with transcripts enriched in stress granules and that stress granule localized transcripts with ac4C are specifically translationally regulated. We also show that ac4C on mRNA can mediate localization of the protein NOP58 to stress granules. Our results suggest that acetylation of mRNA regulates localization of both stress-sensitive transcripts and RNA-binding proteins to stress granules and adds to our understanding of the molecular mechanisms responsible for stress granule formation.**

**Keywords** RNA Acetylation; Stress Granules; Translation; mRNA Localization
**Subject Category** RNA Biology

## Introduction

Stress granules (SG) are membrane-less assemblies of mRNA-protein complexes that arise from mRNAs stuck in translation initiation. RNA-protein complexes are important for their formation and the mechanisms promoting stress granule formation involve both conventional RNA-protein interactions and interactions that encompass intrinsically disordered regions of proteins (Khong et al, 2017; Glauninger et al, 2022). SG have been extensively studied, and it is well-established that they form when translation initiation is limited (Khong et al, 2017) and a variety of roles for SG within the cell have been proposed (Khong et al, 2017). While SG assembly and disassembly can be regulated by various

post-translational modifications (Wang et al, 2023) the impact of RNA modifications on their formation, dispersal and function remains largely unclear (Glauninger et al, 2022).

RNA modifications occur at all RNA species, particularly at rRNA and tRNA, but also mRNA is increasingly reported to contain modified nucleosides (Gilbert et al, 2016). Particularly N6-methyladenosine (m6A) has been shown to play important roles for mRNA translation (Wang et al, 2015). While m6A has been proposed to play a role in mRNA localization to SG through interaction with YTHDF (Fu and Zhuang, 2020; Ries et al, 2023; Anders et al, 2018), a functional relationship has recently been questioned (Khong et al, 2022).

More recently, the RNA modification N4-acetylcytidine (ac4C) has been shown to be deposited on mRNA and regulate translation efficiency (Sas-Chen et al, 2020; Arango et al, 2018, 2022). ac4C is conserved through all kingdoms of life and is induced upon several different stresses (Sas-Chen et al, 2020). ac4C is less abundant than m6A on mRNA and due to difficulties in precise and quantitative mapping its function and occurrence on mRNA has remained controversial (Arango et al, 2018; Sas-Chen et al, 2020). A recent study using nucleotide-resolution non-quantitative mapping of ac4C in HeLa cells identifies more than 6,000 acetylation sites in mRNA, demonstrating a wide-spread occurrence of acetylation on human mRNA (Arango et al, 2022).

Here, we show that ac4C is enriched in SG and that acetylated transcripts are predominantly localized to SG in response to oxidative stress induced with arsenite. We propose that acetylation of RNA can affect mRNA localization to SG, in part by affecting the translational release of mRNA from the ribosome (Arango et al, 2022) providing new insight into both the function and consequences of mRNA acetylation and mechanism of RNA localization to SG.

## Results

### Acetylated RNA is enriched in membrane-less organelles

ac4C is known to occur on 18S rRNA that is highly present in the nucleolus (Bortolin-Cavaillé et al, 2022), where the acetyltransferase NAT10 is also predominantly localized (Larrieu et al, 2018), as well as

[1]Institute of Molecular Biology and Genetics, Aarhus University, 8000 Aarhus, Denmark. [2]Institute of Biomedicine and Translational Medicine, University of Tartu, 50411 Tartu, Estonia. [3]Institute of Biomedicine, Aarhus University, 8000 Aarhus, Denmark. [4]Max Planck Institute for Molecular Genetics, 14195 Berlin, Germany. [5]Proteomics and Modomics Experimental Core (PROMEC), Department of Clinical and Molecular Medicine, Norwegian University of Science and Technology and the Central Norway Regional Health Authority, 7030 Trondheim, Norway. ✉E-mail: ulf.orom@mbg.au.dk

on tRNA and mRNA (Arango et al, 2018). In addition, ac4C is induced by oxidative stress from archaea to mammals (Sas-Chen et al, 2020), suggesting an evolutionary conserved role in regulation of gene expression or mRNA function. In human, ac4C is deposited by the acetyltransferase NAT10 (N-acetyltransferase 10) (Arango et al, 2018), and depletion of NAT10 by targeting exon 5 of the NAT10 gene with CRISPR-Cas9 for inducing mutations in both alleles (NAT10 KO) results in decreased ac4C levels across all RNA species (Fig. EV1A,B).

To assess ac4C distribution in the cell during non-stressed and stressed conditions we used microscopy and staining for nucleolar marker NCL and the SG markers G3BP and PABP combined with oxidative stress induced by arsenite (Figs. 1A,B and EV1A). In addition, we determined the co-localization of ac4C with SG upon osmotic stress induced by sorbitol (Fig. EV1A). Here, we used WT and NAT10 KO HeLa cell lines with the latter showing an 80 per cent decrease of ac4C levels on mRNA (Fig. EV1B) and a decrease of the expression levels of the inactivated NAT10 protein (Fig. EV1C).

In unstressed conditions we see that ac4C localize to nucleoli both in WT and NAT10 KO cells (Fig. 1A,B), albeit with different intensities due to the different baseline levels of ac4C. Upon

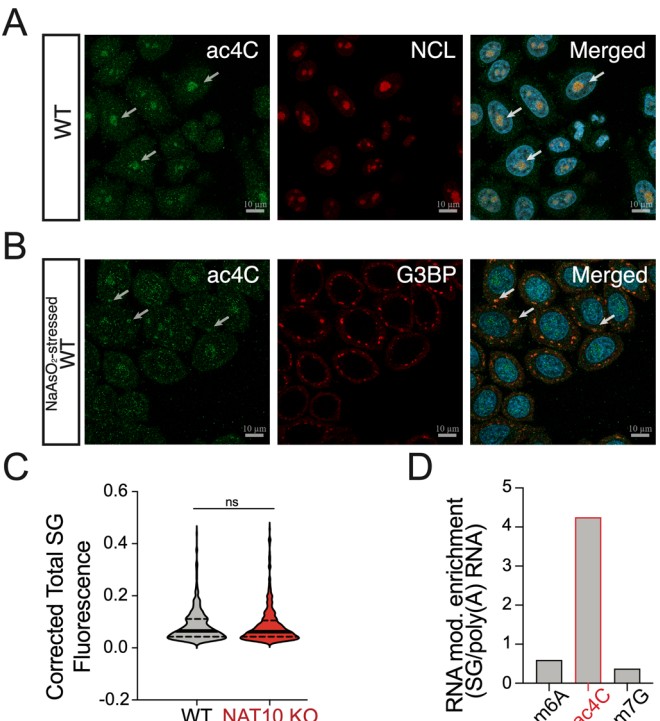

Figure 1. ac4C enrichment in stress granules.

Untreated WT HeLa cells stained for ac4C (green) and the nucleolus marker NCL (Nucleolin; red) (A) and arsenite stressed WT HeLa cells stained for ac4C and the SG marker G3BP (red) (B). DAPI staining (nuclei) is in blue. Arrows indicate ac4C granules overlapping with nucleoli in (A) and SG in (B). Scale bar 10 μm. In (C) the intensities of SG, defined from G3BP fluorescence signal of arsenite-stressed WT and NAT10 KO HeLa cells are quantified and shown as corrected total SG fluorescence. At least 25 cells are analyzed per each condition. Median (solid) and quartiles (dashed), unpaired two-tailed Student's t-test. In (D) is shown the ratio between SG levels of m6A, ac4C and m7G compared to their respective levels on mRNA, measured by RNA mass spectrometry (MS). Source data are available online for this figure.

oxidative stress caused by arsenite or osmotic stress induced by sorbitol we see formation of SG in both WT and NAT10 KO cells as visualized by staining for both the SG markers G3BP and PABP, where particularly arsenite stress mediates ac4C partitioning into SG (Figs. 1B and EV1A), demonstrating a shuttling of acetylated RNA into SG, and suggesting such re-localization of acetylated transcripts to be a general stress response feature. Depletion of NAT10 does not affect the formation nor the size of SG in our experiments, determined by intensity of G3BP staining (Figs. 1C and EV1D), suggesting that ac4C is involved in RNA localization to SG but not essential for SG formation. As an additional control, we treated cells with RNase and repeated staining for ac4C and the SG marker G3BP upon induction of stress with arsenite (Fig. EV1E). The apparent nuclear localization is most likely caused by digested acetylated nucleotides remaining after RNase treatment. Here, we see that the ac4C signal is retained in the nucleus and SG are formed to lesser extent, demonstrating that ac4C is not shuttled out of the nucleus nor into SG, underlining the specificity of the IF signal of ac4C localization to SG upon arsenite stress.

As the high concentration of mRNA in SG could lead to the increased ac4C signal seen with immunofluorescence, we purified RNA from SG as described (Khong et al, 2017) and performed RNA mass spectrometry to quantify the relative abundance of RNA modifications in SG. For comparison, we purified poly(A) RNA, total RNA, 18S rRNA and a tRNA fraction (Fig. EV1F–J), to quantify the modification levels at different RNA species present in SG. In addition to ac4C, for comparison and control of specificity we quantified the common mRNA modifications m6A known to be deposited at mRNA and proposed to be involved in mRNA recruitment to SG (Fu and Zhuang, 2020) and m7G that is present as a 5'cap at mRNAs and also at relatively high levels at tRNA (Fig. EV1J). m6A and m7G are present at different ratios at 18S rRNA and tRNA (Fig. EV1I,J), making it possible to separate the contribution of RNA modification levels by contamination with these RNA species in SG and poly(A) RNA purifications. When we compare SG levels of m6A, ac4C and m7G (Fig. EV1F) to their levels in the mRNA (poly(A) enriched fraction, Fig. EV1G), we see a 4.2-fold enrichment of ac4C in SG whereas m6A and m7G show relative levels of 0.59 and 0.37 fold, respectively (Fig. 1D). This shows that, while a relative enrichment is not reflected in the common mRNA modifications m6A and m7G, we show an enrichment of ac4C, supporting the observations from immuno-fluorescence experiments that ac4C modified RNA are indeed enriched in SG. The profiles of the analyzed RNA modifications in total RNA, 18S rRNA and tRNA, respectively, argue against the observed pattern being a consequence of contamination of the SG purified RNA (Fig. EV1D). Especially the lack of enrichment of m7G, present at relatively high levels in 18S rRNA where ac4C abundance is high (Fig. EV1I), argues against a contamination with rRNA as the reason for increased ac4C in SG.

## The stress granule transcriptome is acetylated

To address the impact of ac4C on the SG transcriptome, we purified and sequenced total RNA from SG with random primer-generated cDNA and without rRNA depletion to obtain the complete picture of the SG RNA content. To purify SG core RNA we used the protocol and controls described in (Khong et al, 2018) with minor modifications as detailed in the methods section (Fig. EV2A). We

sequenced total mRNA (including long ncRNAs) from arsenite stressed cells using random primers after rRNA depletion for comparison to SG to determine relative transcript enrichment (Fig. 2A,B, Dataset EV1). To assess the quality of our SG purification and the reproducibility of the protocol between laboratories and different studies, we compared to two previous studies using the same protocol in U2OS cells (Khong et al, 2017; Matheny et al, 2021). One study used the G3BP as marker for SG purification (Khong et al, 2017), while the other used the SG marker PABP (Matheny et al, 2021). We find a substantial and significant overlap of enriched (Fig. 2C) and depleted (Fig. 2D) transcripts in SG, especially the different cell types taken into account, with reproducible enrichment of the core SG mRNAs reported (Khong et al, 2017), confirming a successful and reproducible purification of SG and contained RNA in our study. For this initial comparison we used a standard cut-off of twofold enrichment and $p < 0.05$. We considered the transcripts overlapping in all three datasets as high confidence and compared

translation efficiency and mRNA length (Khong et al, 2017) for transcripts unique to HeLa cells between these studies and the high confidence set. For the translation efficiency we do not see a significant difference between the two sets (Fig. 2E), whereas for transcript length (Fig. 2F) the mRNAs in the high confidence set are significantly longer on average (by one-way ANOVA). As transcript length is one of the well-described properties directing mRNA to SG, we increased the fold change cut-off to >4-fold to favor inclusion of high confidence transcripts that are identified in SG from different cell lines using different antibodies for purification for the following analysis, to increase general importance and reproducibility across cell lines and protocols.

Comparing transcript levels in SG to total RNA, we find 418 transcripts that are enriched more than 4 times in SG compared to total RNA in WT cells (with a p.adj.<0.05), whereas in NAT10 KO cells only 55 transcripts are enriched using these same criteria (Dataset EV1). Our data suggest that the RNA content of SG in NAT10 KO cells is closer to the average distribution of mRNA in

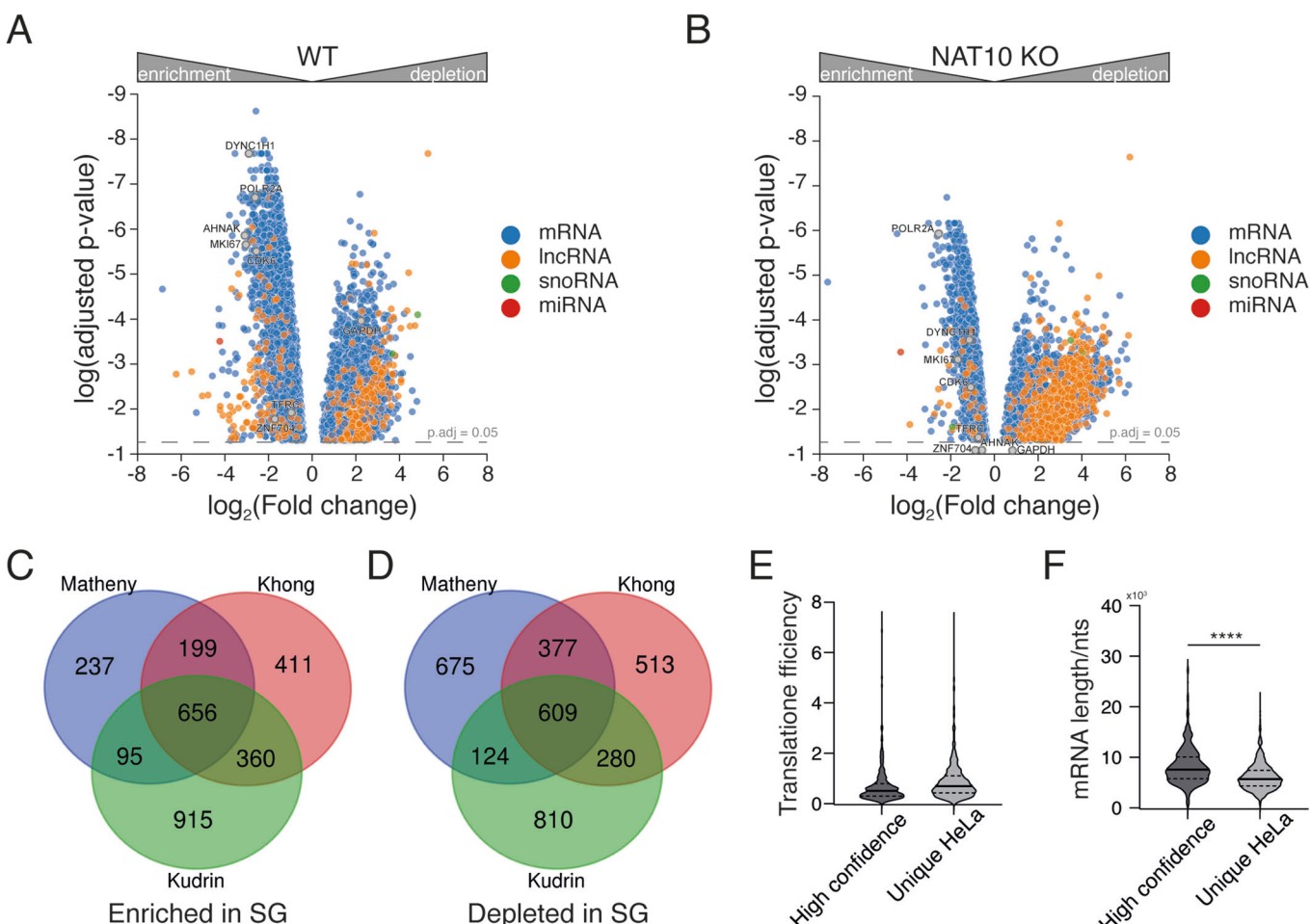

**Figure 2. Purification and sequencing of stress granule-associated RNA.**

Panel (**A, B**) shows volcano plots of SG RNA compared to total RNA for WT (**A**) and NAT10 KO HeLa cells (**B**) from RNA sequencing experiments done in four biological replicates. Benjamini-Hochberg adjusted *p*-value. We compared enriched (**C**) and depleted (**D**) transcripts with previous studies purifying SG from U2OS cells (Matheny et al, 2021; Khong et al, 2017). We assessed translation efficiency (**E**) and mRNA length (**F**) for high-confidence SG transcripts enriched >2-fold in all studies and those enriched only in HeLa cells in the present study, respectively ($n = 559$ high-confidence transcripts, $n = 915$ transcripts unique in HeLa). Unpaired two-tailed Student's *t*-test, ****$p < 0.0001$.

the cell. We used a recently published study with single-nucleotide non-quantitative mapping of ac4C sites in HeLa WT and NAT10 KO cells to compare the possible acetylation sites of mRNAs enriched and depleted in SG, respectively (Arango et al, 2022). When compared to acetylation status using these data (Arango et al, 2022) we find that 223 of the 418 (53%) SG enriched transcripts in WT cells are acetylated (Fig. 3A), which is a significantly higher fraction than expected (3.0 times more than expected, $p < 2.7E{-}63$ by hypergeometric distribution).

Most acetylated mRNAs in the transcriptome are found to have a single site that can be ac4C modified (56.6%), while a subset of transcripts displays several possible acetylation sites, up to 25 for the MKI67 transcript (Arango et al, 2022) (Fig. 3B). 8 transcripts contain more than 15 possible ac4C sites, and 7 of these transcripts are enriched more than 4-fold in WT SG, reflected in the general observation that ac4C modified transcripts tend to localize to SG independently of expression level (Figs. 3C and EV2B). The average number of possible ac4C sites in acetylated SG transcripts is 3.7 compared to 2.0 for the

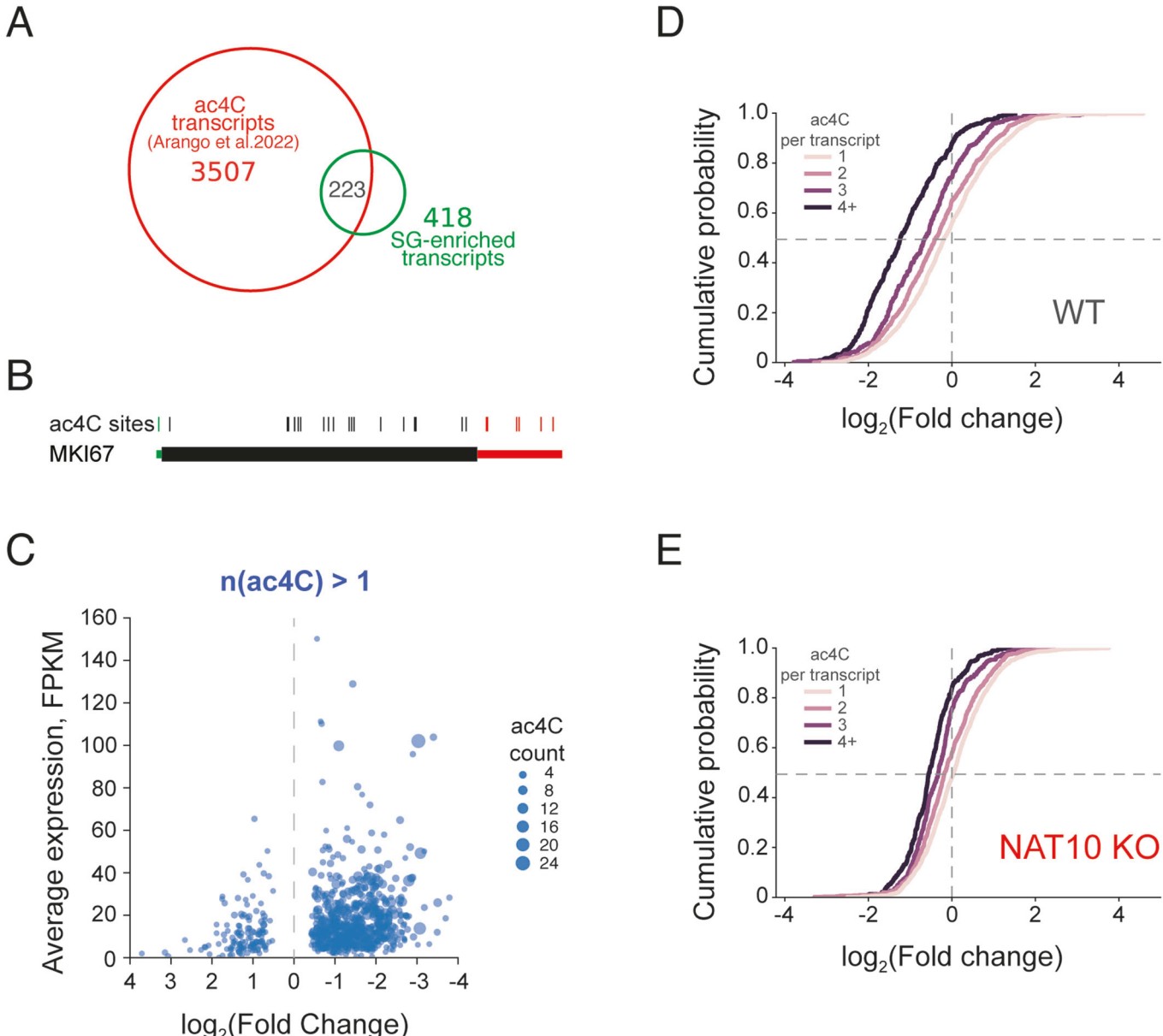

**Figure 3. The ac4C-dependent stress granule transcriptome.**

The overlap between acetylated transcripts in WT HeLa cells and transcripts enriched in SG (53.3 per cent) is shown as a Venn diagram in panel (**A**). The transcript with the most ac4C sites, MKI67, is shown as a schematic in panel (**B**). Panel (**C**) The transcripts with more than one ac4C show tendency towards localization in SG upon arsenite-induced stress in HeLa WT cells. ac4C-transcripts are shown along with fold change enrichment in SG and normalized average expression as FPKM. Panels (**D**) and (**E**) show the cumulative distribution of the log₂ (Fold change) for mRNAs in SG compared with total mRNA. ac4C-containing transcripts are enriched in arsenite-induced SG in HeLa WT cells (**D**), and the effect increases with the number of ac4C sites. With the ac4C depletion (HeLa NAT10 KO cells) (**E**), normally ac4C-enriched transcripts are considerably less abundant in SG compared to WT. The number of mapped ac4C sites for each bin is indicated.

average acetylated transcript in HeLa cells, showing that mRNAs with several ac4C sites are more likely to be localized to SG. We compared the enrichment of acetylated transcripts to SG in WT and NAT10 and analyzed separately for mRNAs with 1, 2, 3 and 4 or more possible ac4C sites. We observe that more acetylated transcripts accumulate more in WT cells, and that the degree of accumulation to SG is decreased around twofold in NAT10 KO cells (Fig. 3D,E), suggesting that ac4C mediates enrichment of acetylated mRNAs in SG.

mRNAs are often reported to localize to SG dependent on their lengths. To assess the contribution of transcript length and ac4C in SG localization, respectively, we determined the impact of length on SG

enrichment in WT cells and NAT10 KO cells with reduced ac4C levels. We binned mRNAs dependent on length and show enrichment compared to total RNA in Fig. 4A,B. We see that mRNA enrichment is correlated with length in WT cells, whereas this enrichment is significantly impaired in NAT10 KO cells, suggesting that length alone is insufficient for strong mRNA localization to SG.

## Translation efficiency is dependent on ac4C levels

Considering the mRNAs with high number of ac4C sites and translation efficiency (TE) as determined by ribosome profiling for

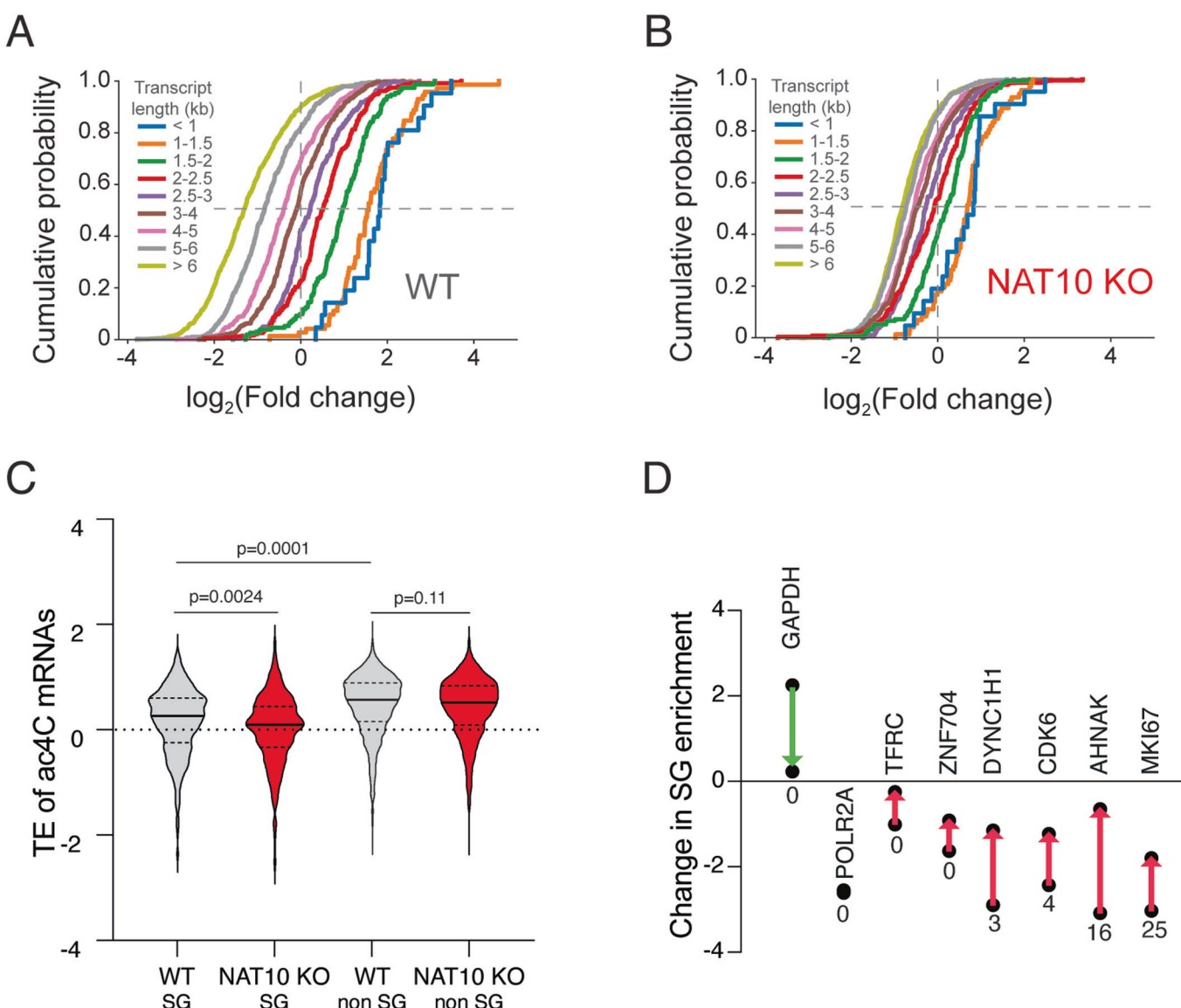

**Figure 4. The contribution of transcript length and translation efficiency to mRNA localization and their dependence on acetylation status.**

(A) mRNA enrichment in SG, compared to total mRNA, is related to transcript length in arsenite-stressed WT HeLa cells. Cumulative distribution of the $\log_2$(Fold change) of transcripts, binned on the basis of length is shown. Compared to WT, transcript length shows considerably weaker influence on mRNA localization to SG in arsenite-stressed ac4C-depleted (NAT10 KO) HeLa cells (B). For the further analysis we used a cut-off of four-fold enriched in SG, and comparison of TE for this set is shown in panel (C) where SG transcripts are defined as having a $\log_2$FC $< -2$, and transcripts not enriched in SG are defined as having a $\log_2$FC $> -1$ ($n = 449$ SG and $n = 1346$ non-SG), one-way ANOVA. In (D) are shown 8 model transcripts used for comprehensive SG studies (Khong et al, 2017) and their change in SG localization from WT to NAT10 cells. The arrow indicates the direction of the change and below each data point is shown the number of acetylation sites for each transcript, respectively.

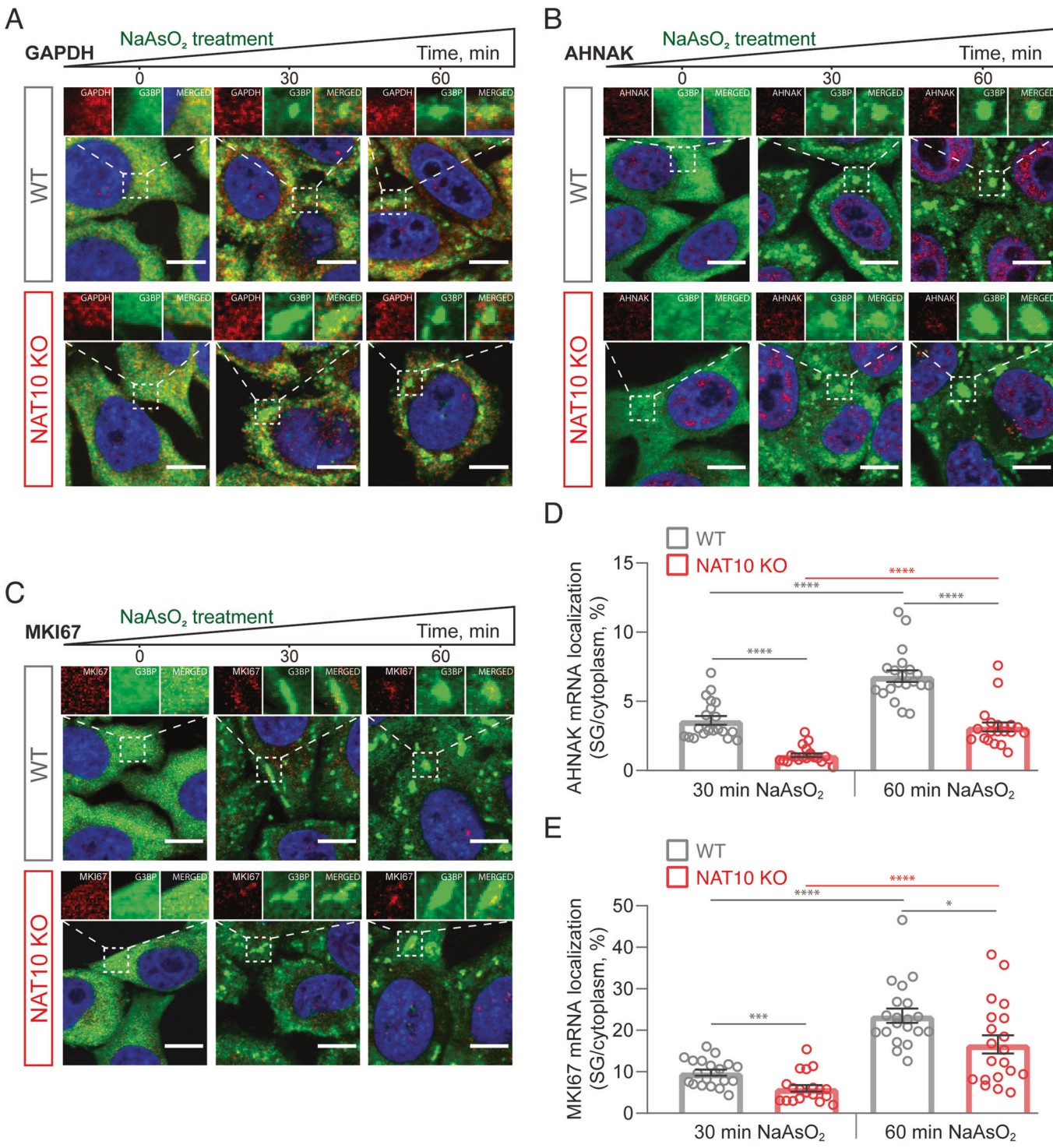

**Figure 5. smFISH analysis of AHNAK, MKI67 and GAPDH mRNA localization to SG.**

We used smFISH, coupled to G3BP-IF, to further substantiate our findings using the control transcript GAPDH (**A**) and core SG transcripts AHNAK (**B**) and MKI67 (**C**). Transcript localization upon arsenite stress for either 0 min, 30 min or 1 h in WT and NAT10 KO HeLa cells is shown. Red is smFISH probe, green is G3BP and blue is DAPI. Scale bar, 10 μm. Quantification of the fraction of AHNAK mRNA (**D**) and MKI67 (**E**) in SG per cytoplasm in different conditions. 20 cells per each condition were counted, each circle represents a single cell. Data is represented as Mean ± SEM, unpaired two-tailed Student's $t$-test, $*p < 0.05$, $**p < 0.01$, $***p < 0.001$ and $****p < 0.0001$. Source data are available online for this figure.

WT and NAT10 KO cells from Arango and colleagues (Arango et al, 2018), we see that acetylated transcripts in SG are less efficiently translated. This is in accordance with the observation that SG are reported to be predominantly composed of long and less efficiently translated transcripts (Khong et al, 2017), that often correlate with lower TE determined by ribosome profiling (Weinberg et al, 2016). When comparing TE of acetylated transcripts enriched in SG compared to non-enriched ones, we see that the SG enriched ac4C transcripts have lower TE on average than acetylated transcripts not enriched in SG ($\log2FC > -1$) (Fig. 4C). In the NAT10 KO cells with reduced ac4C levels, the acetylated transcripts enriched in SG have a lower TE on average than in WT cells. TE for ac4C transcripts not enriched in SG do not appear affected on average, suggesting that translation of transcripts that are prone to accumulate in SG are particularly sensitive to acetylation status, which could apply to long mRNAs often shown to accumulate in SG, which by chance will have more probability of containing an ac4C site.

When we look at ac4C status in HeLa cells of the model transcripts used by Khong et al (Khong et al, 2017), we see that the non-SG control GAPDH is not acetylated. The intermediately SG enriched transcripts POL2RA and TFRC show no acetylation sites as well. For the SG enriched transcripts DYNC1H1, ZNF704, CDK6 and AHNAK all but ZNF704 are acetylated, as well as the top acetylated transcript MKI67 highly enriched in SG (Fig. 4D). The enrichment in SG compared to total RNA of all model transcripts changes between WT cells and NAT10 KO cells towards less pronounced, i.e., SG and total RNA composition becomes more similar in the absence of ac4C, in agreement with the SG transcriptome being less pronounced (Fig. 4D). We observe that those transcripts that are acetylated undergo the largest change in SG enrichment from WT cells to NAT10 KO cells, whereas, e.g., POLR2A that does not contain acetylation sites, and in our data is highly SG enriched, does not change its distribution in the NAT10 KO cells compared to WT cells.

## Acetylated mRNA partitioning to SG is confirmed by smFISH

To further substantiate the localization of acetylated RNA to SG we used smFISH to visualize the localization of control non-SG mRNA GAPDH (Fig. 5A) and the core SG mRNAs AHNAK (Fig. 5B) and MKI67 (Fig. 5C) to SG following arsenite treatment. Here, we see time-dependent partitioning of AHNAK and MKI67 into SG in WT cells, that is significantly diminished in NAT10 KO cells, as quantified in Fig. 5D,E. To further substantiate the identification of SG with the used G3BP protein marker, we co-stained with poly(A) smFISH probes showing a high degree of overlap, as shown in Fig. EV3A. ac4C promotes mRNA decoding efficiency (Arango et al, 2018), and it is thus possible that long mRNAs with ac4C modifications exit translation less efficiently in ac4C depleted NAT10 KO cells. This would result in mRNAs remaining in polysomes and not partitioning into SG. To assess if this is the case we used puromycin treatment at the same time as arsenite treatment, as shown in Fig. EV3B–D. Puromycin releases residual ribosomes that would inhibit partitioning of mRNA into SG (Khong and Parker 2018) and we observe, that puromycin treatment yields more efficient partitioning of both AHNAK and

MKI67 into SG in both WT and NAT10 KO conditions for both 30′ and 1 h timepoints (Fig. EV3E,F), but that the partitioning into SG is still significantly decreased in NAT10 KO cells. The overall distribution of AHNAK and MKI67 into SG show a markedly reduced partitioning into SG in NAT10 KO cells compared to WT (Figs. 5D,E and EV3E,F), and while the puromycin effect is clearly promoting localization of mRNA to SG it does not overrule the impact of ac4C modification of partitioned transcripts. The expression on mRNA level of AHNAK is higher in NAT10 KO cells by RNA sequencing, further supporting that the decreased partitioning of AHNAK into SG in NAT10 KO cells is an effect of ac4C rather than of mRNA levels in the cell. Thus our data suggest that ac4C directly affects the targeting of RNA into SG rather than indirectly through its reported effect on decoding efficiency (Arango et al, 2018).

## Identification of ac4C binding proteins

While ac4C could directly regulate partitioning to SG, interactions with proteins that mediates the localization is also a possibility. To identify protein binders recognizing the ac4C modification, we in vitro synthesized three biotinylated 76 nts RNAs containing 21, 24 and 16 C nucleotides, respectively, in various sequence contexts, including the known ac4C consensus site CCG (Fig. EV4A,B). The RNA was synthesized in the presence of unlabeled A, U and G ribonucleotides and three different concentrations of ac4C ribonucleotides (0, 50 and 100% ac4C compared to C, respectively), and yielded between 25 and 60% incorporation of ac4C into the RNAs (Fig. EV4C,D). The in vitro synthesized RNA was incubated with total cellular lysate from NAT10 KO cells, to increase the unbound fraction of ac4C binders. We eluted bound proteins from RNA using excess biotin and RNase, and subjected purified proteins to liquid-phase mass spectrometry (LC/MS). Here, we identify a list of proteins preferentially bound by the ac4C labeled RNA compared to the control non-acetylated RNA (Dataset EV2).

As a complementary approach to identify the most biological relevant protein interacting with the SG enriched transcripts, we used Catrapid (Livi et al, 2016) to predict proteins interacting with MKI67, that is highly enriched in SG and contains the most ac4C sites of all transcripts in HeLa. Here, we find that NOP58 is the top predicted protein to bind MKI67, which is also one of the best hits of proteins identified to bind acetylated RNA in our mass spectrometry analysis. When predicting interaction propensity for the model transcripts GAPDH, POL2RA, TFRC, DYNC1H1, ZNF704, CDK6, AHNAK and MKI67 used for the analysis in Fig. 4D, we see a high degree of correlation between the interaction propensity score from Catrapid and the change in SG localization between WT and NAT10 KO cells (Fig. 6A), supporting the identification of NOP58 as a protein binding preferentially to transcripts with several ac4C sites.

We focused our further validation and functional characterization of ac4C binding proteins in SG localization on NOP58. NOP58 is a nucleolar protein that has also been shown to be localized to SG (Marmor-Kollet et al, 2020). Of note, in unstressed cells, ac4C is highly present in the nucleolus. We validated the interaction between ac4C modified RNA and NOP58 by repeating the pull-down procedure using RNA with different incorporation of ac4C and subsequent WB of NOP58, confirming a dose-dependent binding between ac4C

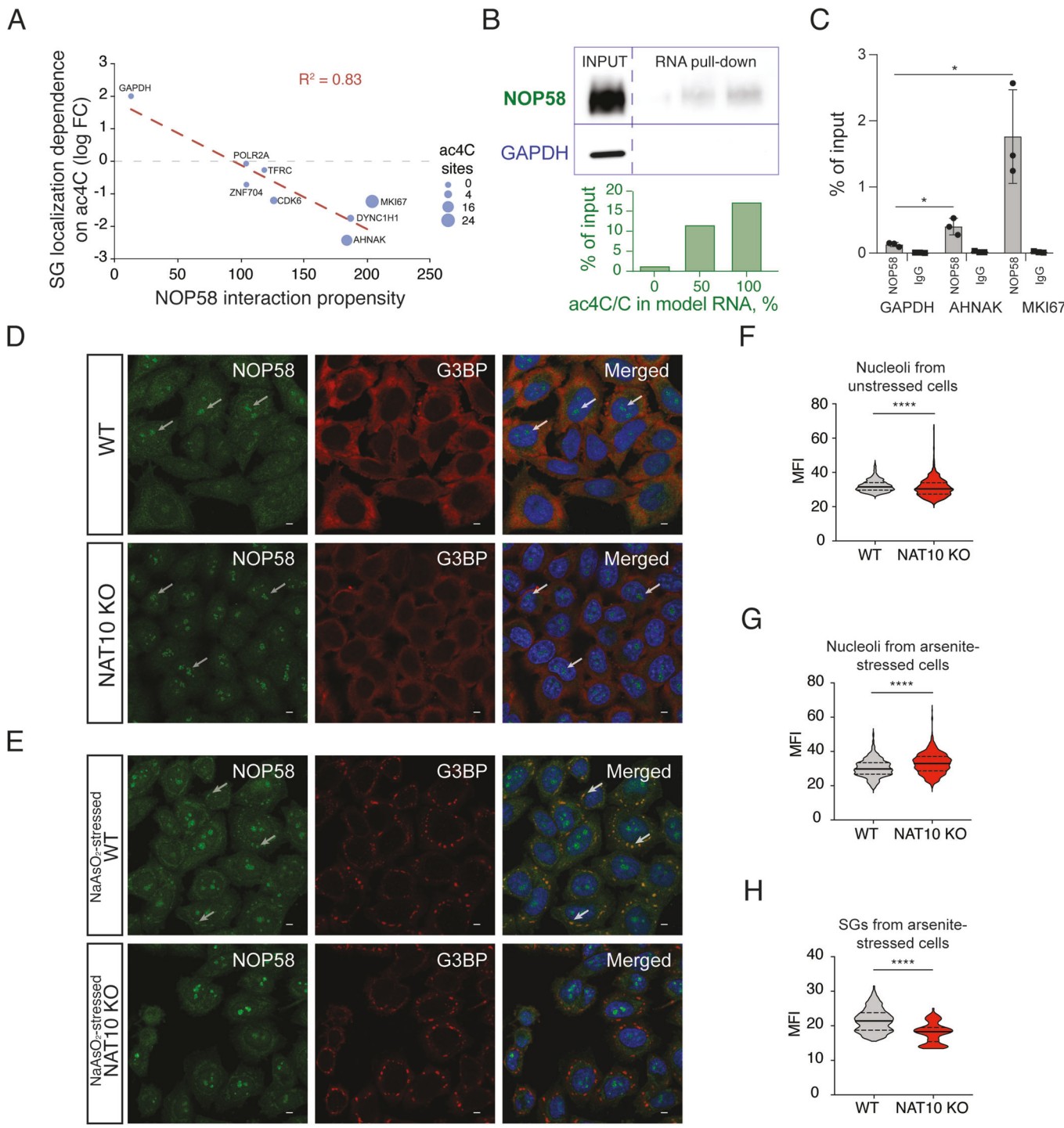

modified RNA and NOP58 protein, whereas no binding is seen to the non-ac4C binding protein GAPDH (Fig. 6B, quantified in lower panel). Using RIP-qPCR, compared to IgG control, we further validated the interaction between NOP58 and the core SG mRNAs AHNAK, having 16 possible ac4C sites, and the most acetylated mRNA MKI67, having 25 possible ac4C sites, whereas GAPDH shows little binding to NOP58 protein (Fig. 6C).

To study the cellular connection between ac4C modified RNA and NOP58 protein localization to SG we stained for NOP58 in WT and

NAT10 KO cells in either unstressed cells or in arsenite-stressed cells. In untreated cells, NOP58 protein localizes to nucleoli in both WT and NAT10 KO HeLa cells (Fig. 6D). Upon arsenite stress and induction of SG we see that in WT HeLa cells NOP58 is recruited to SG (Fig. 6E, upper panel) (in agreement with the SG proteome from (Marmor-Kollet et al, 2020)), whereas in NAT10 KO cells this recruitment is abrogated (Fig. 6E, lower panel). Quantification of NOP58 levels show that NAT10 KO HeLa cells have higher levels of NOP58 in the nucleoli (Fig. 6F,G), and that SG levels of NOP58 are significantly higher in WT

**Figure 6. NOP58 protein localization to SG is dependent on ac4C.**

Panel (**A**) shows the change in SG enrichment from WT to NAT10 KO HeLa cells for the SG model transcripts as a function of interaction propensity with NOP58 (identified as ac4C-binder by LC/MS) predicted by Catrapid. The size of the datapoints show the number of acetylation sites on each transcript. (**B**) Dose-dependent interaction of NOP58 to acetylated RNA oligonucleotides validated by western blot (upper panel) and quantified as compared to 100% input (lower panel). GAPDH is used as a control. (**C**) RIP-qPCR of NOP58 and IgG (as negative control) interactions with the SG core transcript AHNAK as well as MKI67 and GAPDH in WT HeLa cells. Data of three biological replicates is represented as Mean ± SD, unpaired two-tailed Student's *t*-test, *\*p* < 0.05. Immunostaining of NOP58 (green) and G3BP (red) for unstressed cells are shown in panel (**D**) and for arsenite stressed cells shown in panel (**E**). Arrows indicate NOP58 granules overlapping with nucleoli in (**D**) and SG in (**E**). NOP58 containing SG are not indicated in NAT10 KO HeLa cells in (**E**) as they are not visible. DAPI staining (nuclei) is in blue. Scale bar 10 μm. Panels (**F–H**) shows quantification of NOP58 Mean Fluorescence Intensity (MFI) in nucleoli of unstressed (**F**) and arsenite stressed cells (**G**). In (**H**) is shown quantification of MFI of SG in arsenite stressed WT and NAT10 KO HeLa cells. At least 50 cells per each condition were analyzed. Median (solid) and quartiles (dashed), unpaired two-tailed Student's *t*-test, *\*\*\*\*p* < 0.0001. Source data are available online for this figure.

compared to NAT10 KO HeLa cells (Fig. 6H). As NOP58 partitioning to SG is dependent on ac4C, our data suggest that ac4C modified RNA binds NOP58 and is able to partition the protein to SG, proposing that acetylated RNA could play a role in shaping both RNA and protein content of SG.

## Discussion

A recent review on SG asked the question, how does RNA contribute to SG formation, and which RNAs are important? (Glauninger et al, 2022). Several studies using diverse approaches have suggested that transcript length is the key determinant of mRNA recruitment to SG (Khong et al, 2017; Matheny et al, 2021; Namkoong et al, 2018). Here, we show that transcripts modified with ac4C are particularly important for defining the diversity of mRNA in SG. The formation of SG is maintained in NAT10 KO HeLa cells albeit with different mRNA content, showing that SG can still form but the mRNA distribution is more similar to the average mRNA distribution of the cell. We do not see relative enrichment of m6A in SG which could be expected due to the observation that m6A mRNA is transported to SG by YTHDF (Fu and Zhuang, 2020; Ries et al, 2023; Anders et al, 2018), but this lack of enrichment of m6A is in line with a recent study showing a modest impact of m6A on SG RNA composition (Khong et al, 2022).

Our findings that ac4C affects the composition and distinctness of SG mRNA content adds the question how diverse SG content are across cell lines, tissues and external stimuli, and how this is affected by different acetylation patterns across cell lines and tissues. Part of this question might be answered more in-depth once we have a more comprehensive picture of RNA ac4C in a panel of cell lines and tissues, but our comparison with previous SG purification from U2OS cells identifying high-confidence SG transcripts suggest that ac4C levels are partially comparable between cell lines and involved in partitioning of RNA into SG.

It is important to take into consideration the nature of the ac4C mapping data we use for the study that are neither quantitative nor single-transcript resolution. While acetylation patterns might differ between whole-cell lysate and isolated SG we assume in our analysis that the same sites are possible acetylation sites and more sites increases the likelihood of a transcript to be acetylated. This also implies that we do not consider all possible ac4C sites in each transcript acetylated at all times.

Our data also suggest that mRNA is not passively dragged along to SG but are localized there dependent on acetylation status and can mediate protein localization to SG. Why acetylated transcripts that are localized to SG have lower TE and are more susceptible to NAT10 KO than other acetylated transcripts is an outstanding question. It might be associated to the translational status of the transcripts, and possibly indicate that low TE mediate acetylation of transcripts with a subsequent acetylation-dependent enhancement of translation, fitting well with very recent data that ac4C promote translation (Arango et al, 2022).

The observation that transcripts that are prone to accumulate in SG show effect on TE dependent on acetylation status could apply as an additional regulatory level on top of impact of length on mRNA distribution to SG. Acetylation of long mRNAs could enhance the translational inhibition of transcripts in response to stress and ensure an efficient cellular response when quick translation regulation is necessary.

Using puromycin treatment and smFISH we address whether the accumulation of ac4C RNA into SG is due to the rate at which mRNAs exit translation. Here, we see that while puromycin treatment increases the rate of mRNA partitioning into SG it does not override the effect of NAT10 KO mediated decrease in ac4C levels, in support of a model where the presence of ac4C directly affects the partitioning rate of mRNA into SG upon arsenite stress.

Our assessment of the ac4C-binding protein NOP58 and its localization into SG also supports a model where the RNA acetylation and partitioning into SG mediates protein localization, and not vice versa, proposing RNA acetylation as an important factor for defining both the transcriptome and proteome of SG induced by arsenite stress, a finding that should be more thoroughly examined in future work focusing on the RNA-protein interactions in SG formation.

With the findings presented here we show an involvement of ac4C modification in SG partitioning of RNA. Previous work has shown that particularly long mRNAs tend to be localized to SG, and as the most highly acetylated transcripts are also very long it is a central question to separate the contributions on length and ac4C. Our data suggest that ac4C contribute to forming the SG transcriptome by increasing the localization of mRNA, especially for long mRNAs, to SG. Such observation has also been reported for m6A (Ries et al, 2023), albeit with a much lower effect on SG localization than we see for ac4C.

SG share many protein components with neuronal granules, and mutations associated with SG formation have been shown to be

implicated in neurodegenerative diseases such as amyotrophic lateral sclerosis (ALS) and multisystem proteinopathy, where SG-like assemblies form (Dubinski and Vande Velde, 2021). With the presented data on the role of ac4C in SG localization and protein interactions, addressing the impact of RNA acetylation could provide novel insight into neurodegenerative diseases.

# Methods

## Tissue culture and purification of SG

If not stated otherwise, HeLa cells were grown in Dulbecco's Modified Eagle's medium (DMEM; Gibco, #41966-029) supplemented with 10% Fetal Bovine Serum (FBS; Gibco) and 1% Penicillin/Streptomycin (P/S; Gibco) at 37 °C with 5% $CO_2$ until 90% confluency. Cells were collected by trypsinization with 0.05% Trypsin-EDTA (Gibco). When needed, the purification of SG cores was performed as per (Khong et al, 2017, 2018). Briefly, 90% confluent HeLa cells from at least $3 \times 15$ cm cell culture dishes were subjected to oxidative stress by 1 hr treatment with 0.5 mM $NaAsO_2$, followed by flash-freeze in liquid $N_2$ and consecutive lysis by passing 7 times through the 27 G × ½" syringe in SG lysis buffer (50 mM Tris-HCl pH 7.4, 100 mM KOAc, 2 mM MgOAc, 0.5 mM DTT, 50 mg/mL Heparin, 0.5% NP40, 1 mM PMSF, 1:100 PI cocktail, Superase-in (1 U/ml) (Thermo)) and fractionation through centrifugation (Fig. EV2A). Stress granule core enriched fraction was then incubated at 4 °C for at least 3 h, rotating with 5 μg anti-G3BP antibody (#61559, CellSignal) prebound to 60 μl Protein G Dynabeads (Invitrogen). After a series of washes ($3 \times 5$ min in 20 mM Tris-HCl pH 7.4, 200 mM NaCl followed by $1 \times 5$ min in 20 mM Tris-HCl pH 7.4, 500 mM NaCl and $1 \times 2$ min in SG lysis buffer, supplemented with 2 M urea; all buffers were additionally supplemented with Superase-in (1 U/ml) (Thermo)) SG RNA elution from the Dynabeads was performed by Proteinase K (Invitrogen) treatment. Total or SG RNA purification, accompanied with DNase I treatment, was performed using QIAzol and miRNeasy Mini Kit (Qiagen) according to manufacturer's instructions. RNA preparation quality was analyzed with RNA 6000 Pico kit in 2100 Bioanalyzer (Agilent).

## Immunofluorescence staining (IF), RNA FISH and confocal microscopy

HeLa cells were grown on a coverslip and, with 90% confluency reached, were, when needed, subjected to oxidative stress by incubation with 0.5 mM $NaAsO_2$ with or without the presence of 10 μg/ml puromycin for either 30 min or 60 min, or osmotic stress with 0.4 M d-Sorbitol (Merck) for 60 min. Cells were fixed with 4% paraformaldehyde in PBS for 10 min at RT. For IF experiments, permeabilisation and blocking were done with 0.1% Triton-X100 and 0.01% Triton-X100/1% FBS in PBS respectively. Primary antibodies (rabbit anti-ac4C, #ab252215, Abcam; rabbit anti-NOP58, #14409-1-AP, Proteintech; mouse anti-NCL, #87792, CellSignal; mouse anti-G3BP, #ab56574, Abcam; mouse anti-PABP, #ab6125, Abcam) were diluted 1:50 in blocking solution and incubated with the samples for 1 h at RT. Subsequently the samples were incubated with secondary antibodies (anti-rabbit Alexa-Fluor488 conjugated and anti-mouse AlexaFluor647 conjugated) at 1:50 dilutions for 1 h at RT.

For RNase A treatment experiment, sodium arsenite stressed HeLa cells, grown on coverslip were first permeabilized in 2% Tween-20, 1× PBS by gentle shaking for 10 min at RT. Next, 0.2 mg/ml RNase A (Invitrogen) was added in 0.01 M NaOAc, 0.1 M Tris-HCl pH 7.4 buffer and incubated for 20 min at RT. Cells were subsequently fixed with 4% paraformaldehyde in PBS for 10 min at RT.

RNA FISH-IF experiments were performed as follows: after fixation with 4% PFA cells were permeabilized by incubation in 0.1% Triton-X100 in PBS for 10 min at RT followed by incubation pre-hybridization/blocking buffer (2× SSC (saline-sodium citrate), 10% Formamide, 3% BSA, Superase-in (1 U/ml) (Thermo)) for 30 min at RT. Hybridization step occurred in a parafilm-sealed humidity chamber with cells incubated in 125 nM RNA FISH probes (Dataset EV3) and 1:50 mouse anti-G3BP, #ab56574, Abcam - containing hybridization buffer (89% Stellaris RNA FISH Hybridization Buffer (SMF-HB1-10, LGC Biosearch Tech), 1% BSA, 10% Formamide) for 3 h at 37 °C in the dark. After hybridization the samples were briefly washed twice with pre-hybridization buffer (2× SSC, 10% Formamide) followed by incubation with secondary, anti-mouse AlexaFluor488-conjugated, antibody, diluted 1:50 in pre-hybridization/blocking buffer, for 30 min at 37 °C in the dark. This incubation step was repeated twice. Additional washing step with Stellaris RNA FISH Wash Buffer B (SMF-WB1-20, LGC Biosearch Tech) was performed for 5 min at RT before mounting the samples on the glass slides.

Samples were mounted on glass slides using SlowFade Gold Antifade Mountant with DAPI (Invitrogen). Confocal images were captured by a Zeiss LSM 800 Airyscan laser scanning microscope or Olympus FV1200MPE multiphoton laser scanning microscope. Zen 2010 or Olympus FluoView FV1000 acquisition software, respectively, and ImageJ (Fiji) were used for imaging and analysis. Specifically, G3BP and NOP58 IF intensities for SG and nucleoli and SG, respectively, were analyzed through the threshold adjustment and consecutive particle analysis tool. To exclude false positives, smFISH images were analyzed in a semiautomatic manner meaning each cell was analyzed individually with SG selection performed manually and fluorescence intensities quantified with selection measurement tool.

## RNA sequencing and data analysis

Total RNA and SG RNA was sequenced with BGI DNBSEQ sequencing technology (BGI). For total RNA samples were depleted for rRNA (Lexogen RiboCop for HMR v2 kit was used for rRNA depletion) and library generated with random hexamers. For SG, we did not deplete rRNA to maintain the complete picture of SG composition, and generated libraries with random hexamers.

Coverage of the sequenced libraries was ~50 million reads (Q20% > 97.2).

## Quality control

Quality control of all of the fastq files were performed with the help of multiqc (Ewels et al, 2016). Percentage of uniquely mapped reads was in the range of ~91–95%, with ~3–7% of multi-mapped reads.

## Alignment of reads

Alignment of all the paired end reads with the human genome assembly hg19 was performed using STAR (version 2.7.3a) (Dobin et al, 2013), Samtools view (version 1.3.1) (Dobin et al, 2013; Li et al, 2009) command was used to convert bam files to sam files. Further, quantification of the aligned transcripts from the reference hg19 was performed with the help of htseq-count (Anders et al, 2015) using intersection strict as a mode and stranded yes as the parameters.

## Quantification, differential analysis, and annotation

For analysis of RNA expression, readcounts from input samples were used applying a CPM cutoff of 1 or above in all four biological replicates to discriminate expressed genes for the entire dataset. Genes were normalized using the TMM algorithm (Robinson and Oshlack, 2010) and calcNormFactor of edgeR (Oshlack et al, 2010). We next used the relationship function voom (Law et al, 2014) from the limma (Smyth, n.d.) package to establish the mean variance relationship and generate weights for each observation. The lmFit function of limma was used to transform the RNA-Seq data before linear modeling and find differentially expressed (DE) genes.

## In vitro transcription of ac4C modified RNA

T7 promoter containing double stranded DNA (Dataset EV3) was used as a template for in vitro transcription with HighYield T7 mRNA Synthesis Kit (ac4CTP) (Jena Bioscience).

| DNA template variant 1: |
| --- |
| sense strand |
| GTACGGTAATACGACTCACTATAGGGATTGTGCGTGAGATGCACATTC CTGACCGGTGTCTCTTTCTTGACCGGGCCATCCCACATCCGCCGACGC |
| antisense strand |
| GGCCGCGTCGGCGGATGTGGGATGGCCCGGTCAAGAAAGAGACACCG GTCAGGAATGTGCATCTCACGCACAATCCCTATAGTGAGTCGTATTACC |
| DNA template variant 2: |
| sense strand |
| GTACGGTAATACGACTCACTATAGGGAGTGGTCTACACACATGACA GAATGGGGCAGGTCCGTAATCGGTTGCAGAGCGGTTACCGATCTCA TCGC |
| antisense strand |
| GGCCGCGATGAGATCGGTAACCGCTCTGCAACCGATTACGGACCTGCC CCATTCTGTCATGTGTGTAGACCACTCCCTATAGTGAGTCGTATTACC |
| DNA template variant 3: |
| sense strand |
| GTACGGTAATACGACTCACTATAGGGCTTATCTAGTGCATCCGCCGAA ATTACCTGTTGCACGACCACGCTCTGCCGCCTCTCAGACTCCTAACGC |
| antisense strand |
| GGCCGCGTTAGGAGTCTGAGAGGCGGCAGAGCGTGGTCGTGCAACAG GTAATTTCGGCGGATGCACTAGATAAGCCCTATAGTGAGTCGTATTACC |

Either CTP or ac4CTP substrates were used to obtain the certain level of acetylated cytidines within RNA. Purified RNA was subjected

for 3′ end biotinylation with Pierce™ RNA 3' End Biotinylation Kit (Thermo) resulting in 76 nt long RNA of the following sequences:

| Variant 1: |
| --- |
| GGCCGCGUCGGCGGAUGUGGGAUGGCCCGGUCAAGAAAGAGACACC GGUCAGGAAUGUGCAUCUCACGCACAAUC-C(biotin) |
| Variant 2: |
| GGCCGCGAUGAGAUCGGUAACCGCUCUGCAACCGAUUACGGACCUG CCCCAUUCUGUCAUGUGUGUAGACCACUC-C(biotin) |
| Variant 3: |
| GGCCGCGUUAGGAGUCUGAGAGGCGGCAGAGCGUGGUCGUGCAACAG GUAAUUUCGGCGGAUGCACUAGAUAAGC-C(biotin) |

with either 0% or 100% of Cs acetylated.

## Purification of ac4C binding proteins

50 µL of Neutravidine SpeedBeads (Sigma) beads per reaction were equilibrated in buffer A (20 mM Tris-HCl pH 7.4, 1 M NaCl, 1 mM PMSF, PI cocktail, Superase-in (1 U/ml) (Thermo), 1 mM EDTA) followed by addition of 50 pmol of biotinylated model RNA with or without acetylated Cs. After the incubation on rotator at RT for 1 h the beads were washed three times and resuspended in buffer B (20 mM Tris (pH 7.4), 50 mM NaCl, 2 mM MgCl2, 0.1% Tween™-20). During the incubation step the HeLa total cell lysate was prepared by lysing freshly collected HeLa cells in RIPA buffer (Sigma) (1 ml per 15 cm plate) supplemented with 1:100 protease inhibitor cocktail (Sigma) and 1 mM PMSF on ice for 15 min followed by sonication and another 15 min on ice. Cell debris was removed by centrifugation at 4 °C for 10 min at $\geq 10,000 \times g$. Protein concentration was determined by Bradford assay. 100 µg of HeLa lysate per reaction was mixed with 1× buffer B, 15% glycerol and RNase-free $H_2O$ with consecutive addition on RNA-beads mix and incubation at 4 °C for 60 min with rotation. The beads were washed three times with Wash buffer (20 mM Tris (pH 7.4), 10 mM NaCl, 0.1% Tween™-20, 1 mM PMSF, 1:100 PI cocktail) and RNA-bound proteins eluted in 28 µL of elution buffer (1 mM biotin in Wash Buffer and 2 µL RNase) by incubation shaking at 37 °C for 30 min. Eluted proteins were subsequently analyzed by PAGE, Western Blot and MS.

## Proteomics sample preparation and LC-MS/MS instrument settings

Samples were delivered in 1× PBS, 0.01% SDS, the pH was adjusted to 8.5 by adding a final concentration of 100 mM Tris, followed by denaturing at 95 °C for 10 min at 1000 rpm. 4 µg protein of each sample was further processed. Reduction of cysteines was carried out by adding 1.1 µl of 0.1 M tris(2-carboxyethyl)phosphine at 37 °C for 30 min at 800 rpm, alkylation of cysteines similarly by adding 2.5 µl of 0.2 M 2-chloroacetamide. Samples were digested by trypsin (enzyme-protein ratio 1:40) at 37 °C overnight, desalted and reconstituted in 2% formic acid and 5% acetonitrile in water prior to injection to nano-LC-MS. For each sample, 1 and 3 µg protein were injected. LC-MS/MS was carried out by nanoflow reverse phase liquid chromatography (Dionex Ultimate 3000, Thermo Scientific, Waltham, MA) coupled online to a Q-Exactive HF

Orbitrap mass spectrometer (Thermo Scientific, Waltham, MA), as reported previously (Gielisch and Meierhofer, 2015). Raw MS data were processed with MaxQuant software v1.6.10.43 (Cox and Mann, 2008), runs from the same samples were combined and searched against the human UniProtKB with 75,074 entries, released in 05/2020.

## Validation of ac4C binding proteins and western blot

The eluate was subjected to SDS PAGE on Novex Tris-Glycine 4–20% (Invitrogen) gel followed by either silver staining with Pierce Silver Stain Kit (Thermo) or Western blotting against anti-NOP58 (#ab236724, Abcam) and anti-GAPDH (#5174s, CellSignal) antibodies. To confirm NAT10 KO in CRISPR/Cas9 generated HeLa NAT10 KO cell-line, anti-NAT10 (#13365-1-AP, Proteintech) antibody was used. Western blot was developed using Pierce ECL Western Blotting Substrate (Thermo) and imaged with Amersham Imager 600 (GE Healthcare).

## Size-exclusion chromatography of total RNA

Total RNA was fractionated into tRNA, 18S rRNA and 28S rRNA using two dimensions of size-exclusion chromatography (SEC) carried out on an Agilent HP1200 HPLC system with UV detector and fraction collector. The 1st SEC dimension was performed using a Bio SEC-5 1000 Å, 5 μm, 7.8 × 300 mm column (Agilent Technologies, Foster City, CA) and isocratic elution with 100 mM ammonium acetate (pH 7.0) at 500 μl/min for 40 min at 60 °C, collecting three fractions containing 28S rRNA, 18S rRNA, and RNAs below 200 nt ('small RNAs'), respectively. The fractions were lyophilized and the small RNA fraction was reconstituted in 20 μl of water and subjected to a 2nd dimension of SEC using an AdvanceBio SEC 120 Å, 1.9 μm, 4.6 × 300 mm column (Agilent Technologies, Foster City, CA) and isocratic elution with 100 mM ammonium acetate (pH 7.0) run at 150 μl/min for 40 min at 40 °C.

## Analysis of isolated RNA species using LC-MS/MS

RNA was hydrolyzed to ribonucleosides by 20 U benzonase (Santa Cruz Biotech) and 0.2 U nuclease P1 (Sigma-Aldrich, Saint-Louis, MO) in 10 mM ammonium acetate pH 6.0 and 1 mM magnesium chloride at 40 °C for 1 h, then added ammonium bicarbonate to 50 mM, 0.002 U phosphodiesterase I and 0.1 U alkaline phosphatase (Sigma-Aldrich, Saint-Louis, MO) and incubated further at 37 °C for 1 h. The hydrolysates were added 3 volumes of acetonitrile and centrifuged (16,000 × $g$, 30 min, 4 °C). The supernatants were lyophilized and dissolved in 50 μl water for LC-MS/MS analysis of modified and unmodified ribonucleosides. Chromatographic separation was performed using an Agilent 1290 Infinity II UHPLC system with an ZORBAX RRHD Eclipse Plus C18 150 × 2.1 mm ID (1.8 μm) column protected with an ZORBAX RRHD Eclipse Plus C18 5 × 2.1 mm ID (1.8 μm) guard column (Agilent Technologies, Foster City, CA). The mobile phase consisted of water and methanol (both added 0.1% formic acid) run at 0.23 ml/min, for modifications starting with 5% methanol for 0.5 min followed by a 2.5 min gradient of 5–15% methanol, a 3 min gradient of 15–95% methanol and 4 min re-equilibration with 5% methanol. A portion of each sample was diluted for the analysis of unmodified ribonucleosides which was chromatographed isocratically with 20% methanol. Mass spectrometric detection was performed using an Agilent 6495 Triple Quadrupole

system with electrospray ionization, monitoring the mass transitions 268.1–136.1 (A), 284.1–152.1 (G), 244.1–112.1 (C), 245.1–113.1 (U), 286.1–154.1 (ac$^4$C), 282.1–150.1 (m$^6$A) and 298.1–166.1 (m$^7$G) in positive ionization mode.

## NOP58 binding RNA immunoprecipitation

40 μl of Protein G Dynabeads (Invitrogen) in IP buffer (150 mM NaCl, 10 mM Tris-HCl, pH 7.5, 0.1% IGEPAL CA-630, 1 mM PMSF, 1:100 PI cocktail, Superase-in (1 U/ml) (Thermo) in nuclease free H$_2$O) were tumbled with 5 μg anti-NOP58 antibody (#14409-1-AP, Proteintech) or 5 μg anti-IgG antibody (#30000-0-AP, Proteintech) at 4 °C for at least 6 h. Upon freshly prepared (in RIPA (Sigma) buffer) HeLa cell lysate addition, lysate-antibody-beads mixture in IP buffer was incubated ON at 4 °C with gentle rotation in a final volume of 0.8 mL in protein low-binding tubes. For elution, the beads were resuspended in 1× Proteinase K buffer (100 mM Tris-HCl pH 7.5, 150 mM NaCl, 12.5 mM EDTA, 2% SDS and 120 μg/ml Proteinase K (Invitrogen)) and incubated 1 hr with continuous shaking (1200 rpm) at 37 °C. Magnetic separation rack was applied to collect the supernatant. TRIzol/chloroform treatment was applied to supernatant with consecutive centrifugation at >13,500 rpm for 15 min at 4 °C. Upper phase was collected and 700 μl of RLT buffer and 1400 μl of 100% ethanol were added and mixed thoroughly. The mixture was transferred to an RNeasy MiniElute spin column (QIAGEN) and centrifuged at >12,000 rpm at 4 °C for 1 min. This step was repeated until all sample was loaded to the column. The spin column membrane was washed with 500 μl of RPE buffer once, then 500 μl of 80% ethanol once and centrifuged at full speed for 5 min at 4 °C remove the residual ethanol. RNA was eluted with 14 μl ultrapure H$_2$O. RNA concentration was measured using the Qubit RNA HS Assay Kit as per the manufacturer's instructions.

## qPCR and RNA purification

To validate stress granule enriched transcripts from previously purified SG RNA by qPCR, cDNA was synthesized with RevertAid First Strand cDNA Synthesis Kit (Thermo) according to manufacturer's protocol. Platinum™ SYBR™ Green qPCR SuperMix-UDG (Thermo) and the following primers were used for qPCR:

AHNAK FW 5′TCTTCAGCTCCTGCAGCTCT3′ AHNAK RV 5′CTCCATCTTCCGACTTCAGC3′

MKI67 FW 5′AGCCCCAACCAAAAGAAAGT3′ MKI67 RV 5′TTTGTGCCTTCACTTCCACA3′

GAPDH FW 5′GGTGAAGGTCGGAGTCAACGGATTT3′ GAPDH RV 5′ACCAGAGTTAAAAGCAGCCCTGGTG3′

## Statistics

All statistics are done using unpaired two-tailed Student's *T*-test, unless otherwise stated.

# Data availability

RNA sequencing data from total RNA and SG have been deposited to GEO with accession number GSE212380 (Reviewer token gtuxywqexnotlmn). Riboseq data are available from Arango et al,

2018 with accession number GSE102113 and position of ac4C on HeLa mRNA are from Arango et al, 2022 with accession number GSE162043.

## Peer review information

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

## Acknowledgements

We thank Ana Rebane for providing research facilities for PK during parts of the work for the manuscript. We thank Beata Lukaszewska-McGreal for proteome sample preparation. We thank Shalini Oberdoerffer for kindly providing the NAT10 KO and the WT HeLa cell lines. The LC-MS/MS analyses of RNA were performed by the Proteomics and Modomics Experimental Core (PROMEC), Norwegian University of Science and Technology (NTNU) and The Central Norway Regional Health Authority. This facility is a member of the National Network of Advanced Proteomics Infrastructure (NAPI), which is funded by the Research Council of Norway. We acknowledge AU Health Bioimaging Core Facility for the use of equipment and support of the imaging facility. Work in the author's lab is funded by the Novo Nordisk Foundation, The Lundbeck Foundation, Danish Cancer Society, Independent Research Fund Denmark, The Carlsberg Foundation to UAVØ, Estonian Research Council to PK and the Max Planck Society to DM.

## Author contributions

**Pavel Kudrin**: Conceptualization; Data curation; Formal analysis; Validation; Visualization; Methodology; Writing—original draft; Writing—review and editing. **Ankita Singh**: Data curation; Formal analysis; Visualization. **David Meierhofer**: Data curation; Formal analysis; Visualization; Methodology. **Anna Kuśnierczyk**: Data curation; Formal analysis; Methodology. **Ulf Andersson Vang Ørom**: Conceptualization; Formal analysis; Supervision; Funding acquisition; Investigation; Visualization; Methodology; Writing—original draft; Project administration; Writing—review and editing.

## Disclosure and competing interests statement

The authors declare no competing interests.

# Expanded View Figures

**Figure EV1.  Determination of ac4C RNA modification level in stressed HeLa cells.**

(**A**) ac4C-rich RNAs localize in G3BP- and PABP-containing stress granules (SG) in response to oxidative and osmotic stresses, caused by sodium arsenite and d-sorbitol treatments, respectively. Individual SG are indicated by arrows. For panels (**A,D,E**) ac4C is depicted in green, PABP, G3BP or NCL in red, DAPI staining (nuclei) is in blue, scale bar 10 μm; (**B**) RNA mass spectrometry (RNA MS) analysis shows strong decrease in ac4C level in NAT10 KO HeLa cells. $n = 3$ biological replicates, data represented as Mean ± SD. Unpaired two-tailed *t*-test, ***$p < 0.001$; (**C**) Reduction in NAT10 protein level is confirmed by western blot; (**D**) Decreased ac4C IF signal in NAT10 KO cells suggests decreased ac4C levels in nucleoli of untreated and SG of arsenite-stressed cells. Individual organelles are indicated by arrows; (**E**) ac4C IF signal disappears from SG of arsenite-stressed cells in response to RNase A treatment. The levels of RNA modifications m6a, ac4C and m7G in (**F**) SG, (**G**) poly(A) RNA, (**H**) total RNA, (**I**) 18S rRNA and (**J**) tRNA from WT HeLa cells measured by RNA MS. Experiments are done in 2 replicates for panel (**G**) and three replicates for panels (**F, H–J**), data represented as Mean ± SD.

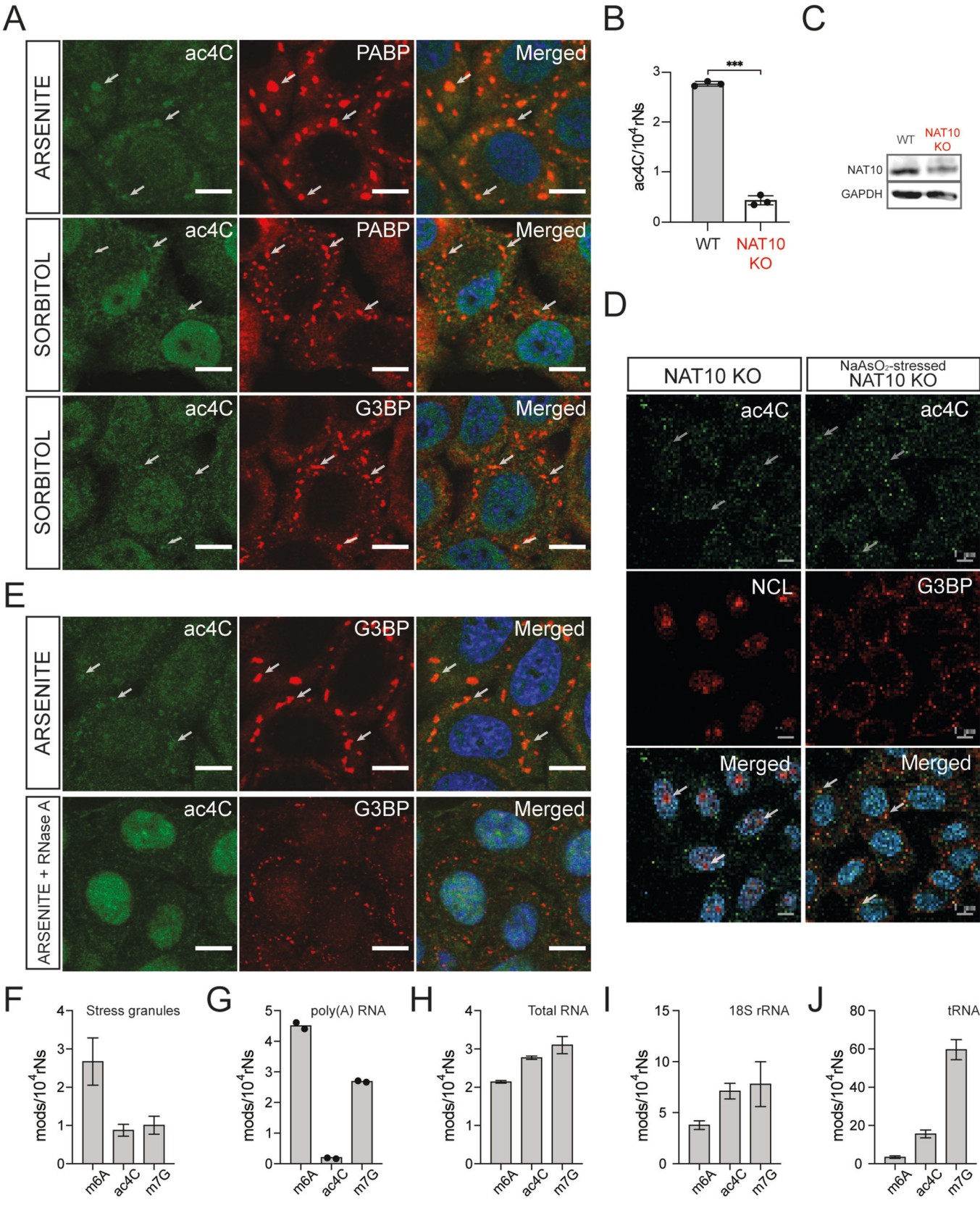

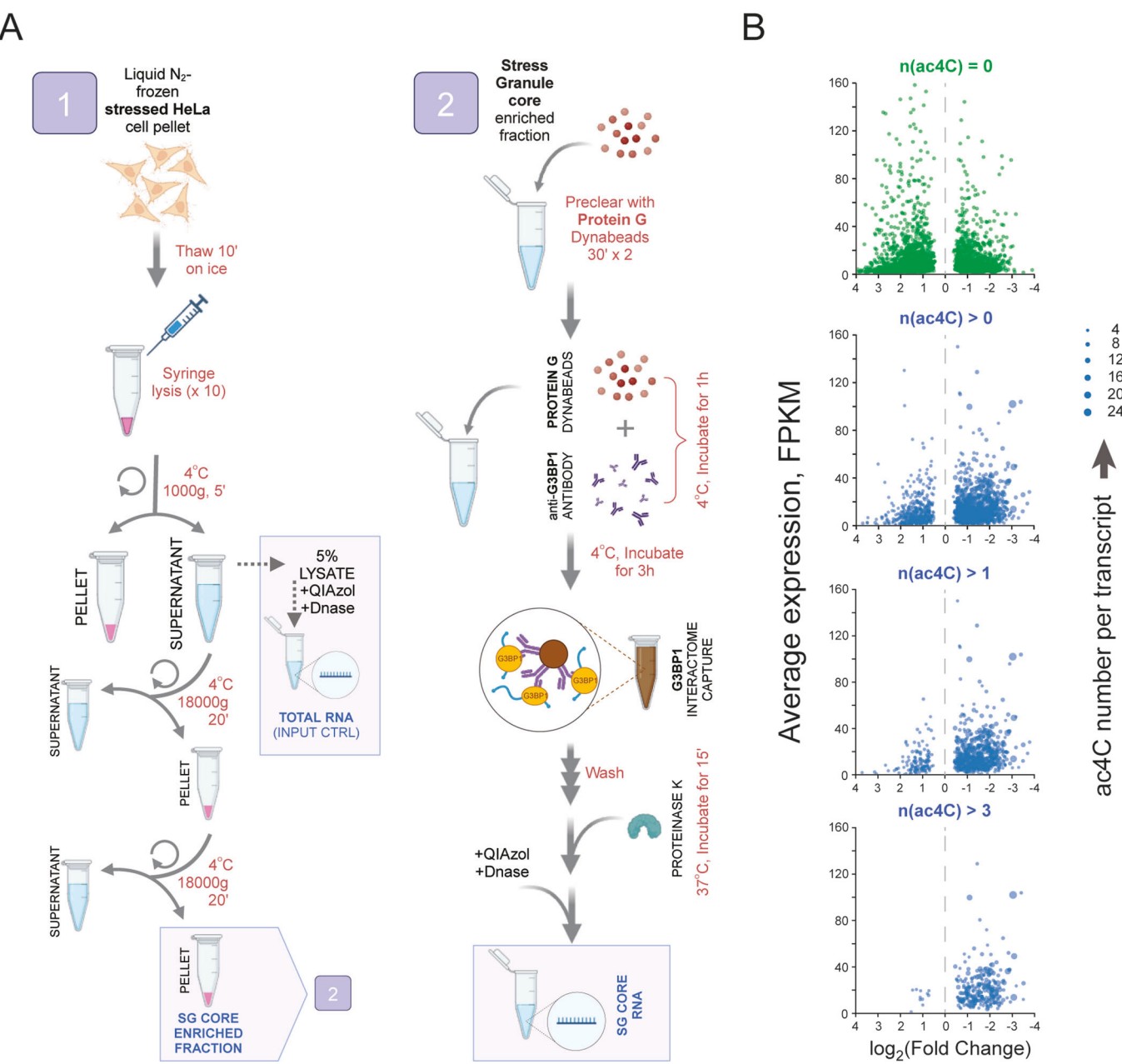

**Figure EV2. Enrichment in SG, normalized expression and ac4C counts on transcripts.**

(A) Schematic representation of SG RNA purification procedure; (B) The figure shows the transcripts with no ac4C sites ($n = 2983$ transcripts), more than zero ac4C sites ($n = 1792$ transcripts), more than one ac4C sites ($n = 865$ transcripts), and more than three ac4C sites ($n = 283$ transcripts) and the found log$_2$FC between SG and total RNA (negative FC means enriched in SG) as well as average normalized expression from RNA-seq shown as FPKM. RNA sequencing experiments were performed in 4 biological replicates.

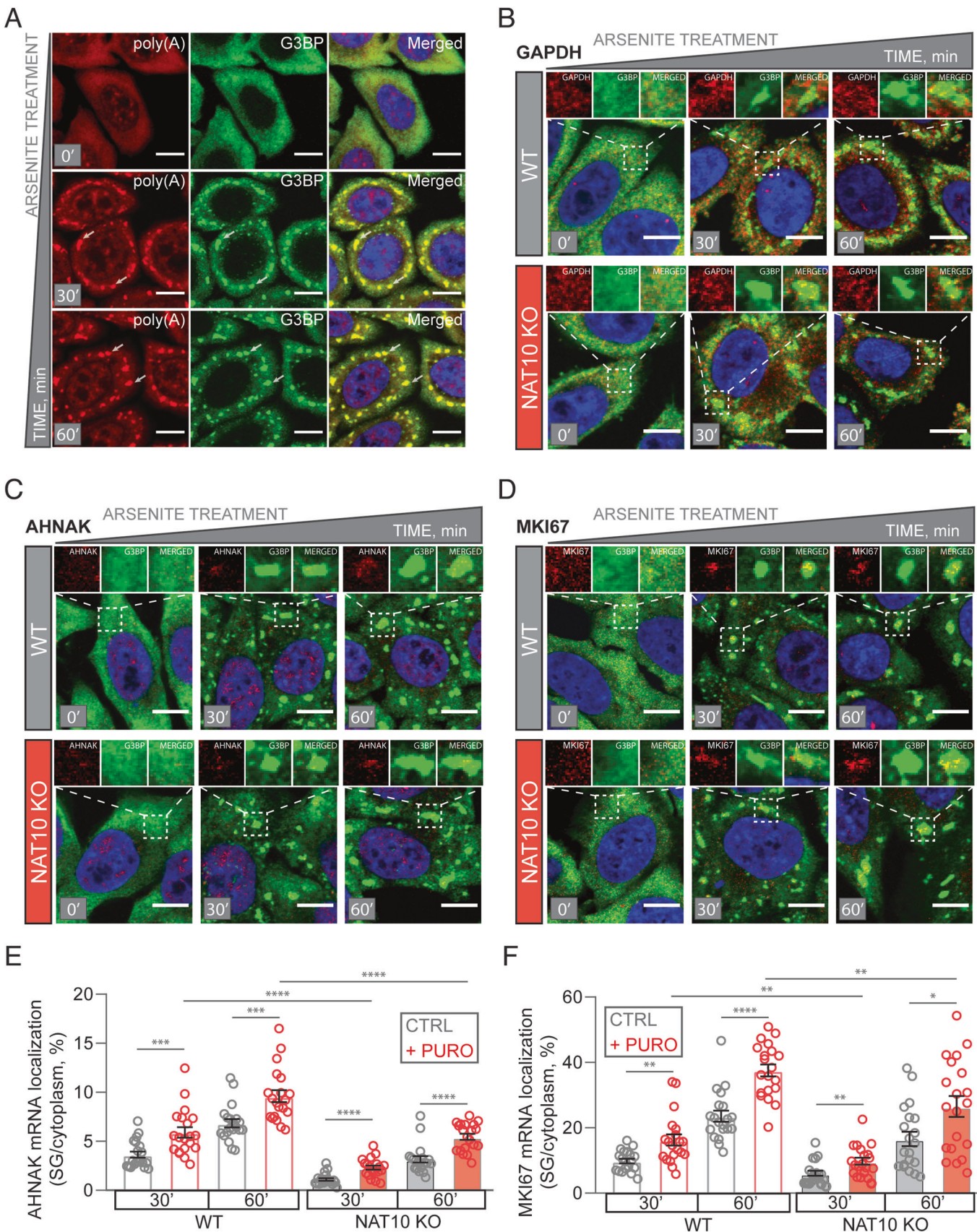

◄ **Figure EV3. mRNA localization to SG in WT and NAT10 KO HeLa cells in response to sodium arsenite treatment.**

(**A**) smFISH-IF microscopy images of poly(A) RNA and G3BP accumulation into SG of WT HeLa cells in response to treatment with 0.5 mM $NaAsO_2$ for 0 min, 30 min or 60 min. Arrows indicate individual SG. Poly(A) RNA is depicted in red (as well as GAPDH mRNA in (**B**), AHNAK mRNA in (**C**) and MKI67 mRNA in (**D**)). For panels (**A–D**) G3BP in green, DAPI staining (nuclei) is in blue, scale bar 10 µm; smFISH-IF microscopy images of (**B**) GAPDH, (**C**) AHNAK and (**D**) MKI67 mRNA localization in response to treatment with sodium arsenite with 10 µg/ml puromycin for 0 min, 30 min or 60 min in HeLa WT and NAT10 KO cells; Quantification of the fraction of AHNAK mRNA (**E**) and MKI67 (**F**) in SG per cytoplasm in different conditions. 20 cells per each condition were counted. Data represented as Mean ± SEM. Unpaired two-tailed Student's *t*-test, *$p < 0.05$, **$p < 0.01$, ***$p < 0.001$ and ****$p < 0.0001$.

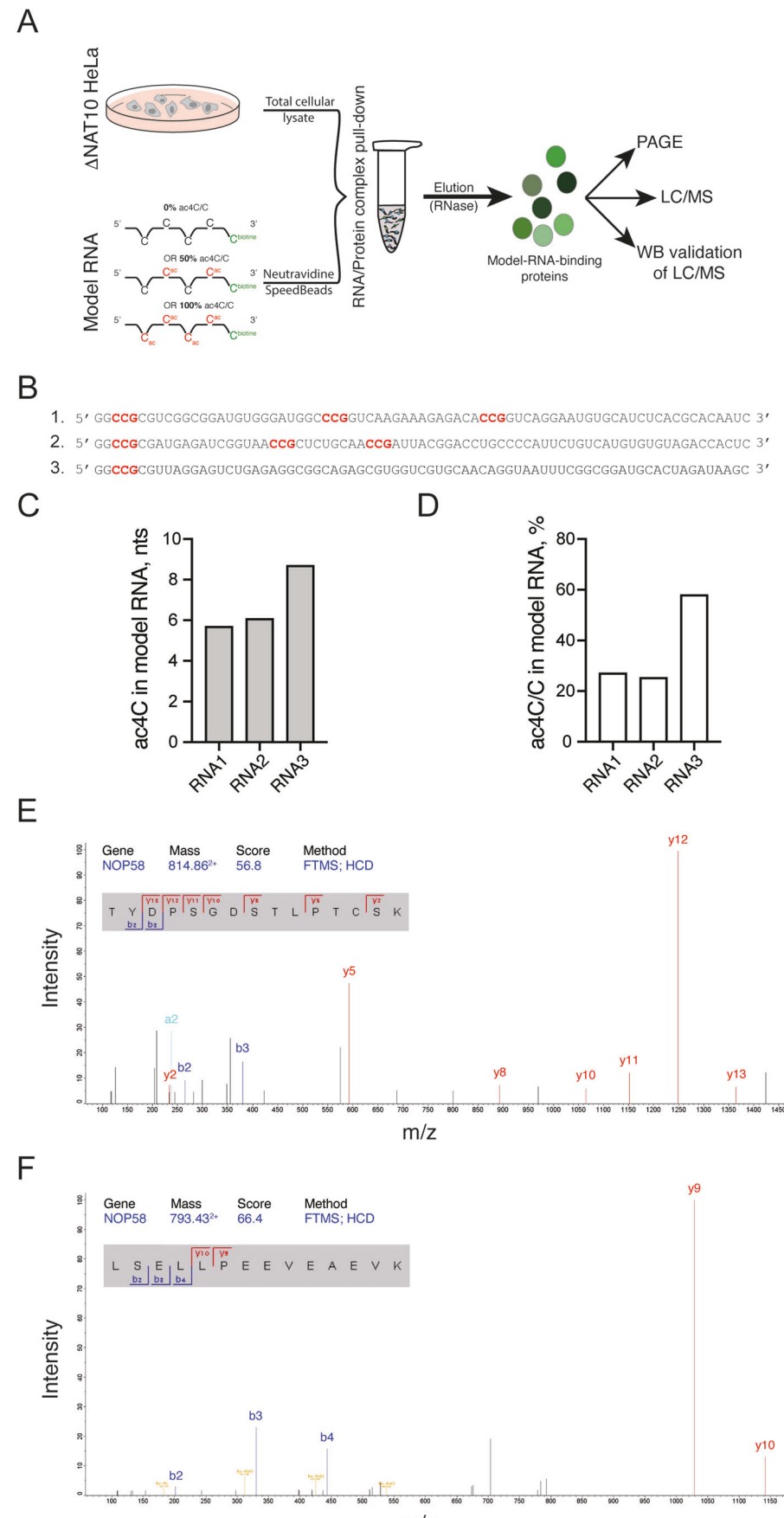

**Figure EV4.   Mass spectrometry identification of NOP58 as an ac4C-binding protein.**

The experimental strategy to identify ac4C-binding proteins is shown in panel (**A**). Panel (**B**) shows the sequences of three RNA oligos used in the immunoprecipitation experiments. Putative NAT10 recognition motifs (CCG) are highlighted in red. Panels (**C** and **D**) show the RNA MS analysis of oligos, where input ac4C/C ratio was 50% during IVT (in vitro transcription) and reflect on actual amount of ac4C in these oligos. (**E**) and (**F**) show peptides used to identify NOP58 in mass spectrometry experiments.

