## [Peer Review File · EMBO Reports]

N4-acetylcytidine (ac4C) promotes mRNA localization to stress granules

Pavel Kudrin, Ankita Singh, David Meierhofer, Anna Kuśnierczyk, and Ulf Orom

Corresponding author(s): Ulf Orom (ulf.orom@mbg.au.dk)

Review Timeline:

Submission Date:	15th May 23
Editorial Decision:	13th Jul 23
Revision Received:	8th Dec 23
Editorial Decision:	1st Feb 24
Editor's Email to the Authors:	5th Feb 24
Revision Received:	6th Feb 24
Accepted:	7th Feb 24

Editor: Esther Schnapp

Transaction Report:

Dear Ulf,

Thank you for the submission of your manuscript to EMBO reports, and I am sorry for my delayed response. We have finally received the full set of referee reports that is pasted below.

As you will see, the referees acknowledge that the findings are potentially interesting and novel. However, they also all have several and overlapping concerns that should be addressed to strengthen the study. For example, the link between ac4C modified mRNAs and SG needs to be strengthened, and missing controls, experimental details and analyses, and complementary strategies should be added. I think all referee concerns are valid and should be addressed, but please let me know in case you disagree, and we can discuss this further, also in a video chat, if you like.

I would thus like to invite you to revise your manuscript with the understanding that the referee concerns must be fully addressed and their suggestions taken on board. Please address all referee concerns in a complete point-by-point response. Acceptance of the manuscript will depend on a positive outcome of a second round of review. It is EMBO reports policy to allow a single round of major revision only and acceptance or rejection of the manuscript will therefore depend on the completeness of your responses included in the next, final version of the manuscript.

We realize that it is difficult to revise to a specific deadline. In the interest of protecting the conceptual advance provided by the work, we recommend a revision within 3 months (13th Oct 2023). Please discuss the revision progress ahead of this time with the editor if you require more time to complete the revisions.

- 1) A data availability section providing access to data deposited in public databases is missing. If you have not deposited any data, please add a sentence to the data availability section that explains that.
- 2) Your manuscript contains statistics and error bars based on $n=2$. Please use scatter blots in these cases. No statistics should be calculated if $n=2$.

5) a complete author checklist, which you can download from our author guidelines

<<https://www.embopress.org/page/journal/14693178/authorguide>>. Please insert information in the checklist that is also reflected in the manuscript. The completed author checklist will also be part of the RPF.

6) Please note that all corresponding authors are required to supply an ORCID ID for their name upon submission of a revised manuscript (<<https://orcid.org/>>). Please find instructions on how to link your ORCID ID to your account in our manuscript tracking system in our Author guidelines <<https://www.embopress.org/page/journal/14693178/authorguide#authorshipguidelines>>

7) Before submitting your revision, primary datasets produced in this study need to be deposited in an appropriate public database (see <https://www.embopress.org/page/journal/14693178/authorguide#datadeposition>). Please remember to provide a reviewer password if the datasets are not yet public. The accession numbers and database should be listed in a formal "Data Availability" section placed after Materials & Method (see also <https://www.embopress.org/page/journal/14693178/authorguide#datadeposition>). Please note that the Data Availability Section is restricted to new primary data that are part of this study. * Note - All links should resolve to a page where the data can be accessed. *
If your study has not produced novel datasets, please mention this fact in the Data Availability Section.

12) All Materials and Methods need to be described in the main text. We would encourage you to use 'Structured Methods', our new Materials and Methods format. According to this format, the Materials and Methods section should include a Reagents and Tools Table (listing key reagents, experimental models, software and relevant equipment and including their sources and relevant identifiers) followed by a Methods and Protocols section in which we encourage the authors to describe their methods using a step-by-step protocol format with bullet points, to facilitate the adoption of the methodologies across labs. More information on how to adhere to this format as well as downloadable templates (.doc or .xls) for the Reagents and Tools Table can be found in our author guidelines: <<https://www.embopress.org/page/journal/17444292/authorguide#textformat>>. An example of a Method paper with Structured Methods can be found here: <<https://www.embopress.org/doi/10.15252/msb.20178071>>.

13) We would like to alert you that EMBO Press offers a new format for a video-synopsis of work published with us, which essentially is a short, author-generated film explaining the core findings in hand drawings, and, as we believe, can be very useful to increase visibility of the work. This has proven to offer a nice opportunity for exposure i.p. for the first author(s) of the study. Please see the following link for representative examples and their integration into the article web page:

<https://www.embopress.org/doi/full/10.15252/embj.2019103932>

I look forward to seeing a revised form of your manuscript when it is ready.

Referee #1:

This manuscript by Kudrin and colleagues reports on a potential relationship between stress granules and the presence of ac4C in mRNA. As little is known of how mRNAs are localized to discrete cellular compartments, the questions pursued are of inherent significance. Through a combination of IF, RNA-seq, mass spec and transcript-specific analyses, the authors conclude that mRNAs are targeted to SGs in a NAT10-dependent manner. Though an appealing model in principle, substantial data issues cast doubt on the directness of the shown associations and overall relevance, as described in the below comments.

Major comments:

Figure 1. (a-d) The provided IF images are of insufficient quality to support the authors' claims. Additional controls are required to ensure that the ac4C staining is specific, including an RNase(+) condition. In addition, quantification of % ac4C overlap in SGs as compared to background should be examined. As presented, it appears nucleoli are disrupted in response to sodium arsenite, which could create the perception of greater SG overlap. Much higher resolution pictures should be provided to better demonstrate co-localization. (f) It is difficult to assess the relevance of these results in the absence of any data showing the efficacy of SG RNA enrichment. It is also unclear what the plot is showing. Are these mass spec results? If so, how are the enrichment values calculated and normalized?

Supplementary Fig. 1. The relative enrichment of ac4C in RNA purified from SGs is intriguing, but it is unclear how the mass spec is normalized. Additional information including controls should be included to ensure sample purity.

Figure 2. The overall relevance of this figure is unclear. (c-d) I don't follow the rationale for examining SG RNAs in different cells. The presented results suggest that the transcripts that are found in SGs vary substantially between cell types, which agrees with my understanding. (e) It has been reported by others that SGs are transient and that their composition can vary substantially in different conditions. With this in mind, the use of TE results that were performed by another lab at another time does not seem appropriate. In addition, ac4C is believed to be a low stoichiometry modification and there is no way of knowing whether the TEs that were measured in bulk relate to the modified pool of mRNA. Relatedly, as the authors have not directly mapped ac4C in SG mRNA, they cannot be confident whether the mRNAs they are examining are in fact acetylated.

Figure 3. (b) As the authors did not map ac4C in SGs, they cannot reasonably conclude that the mRNAs that they detect in SGs are in fact ac4C modified. Though the mass spec shown in Supplemental Figure 1 suggests that some SG resident RNAs are acetylated, the presented data do not establish a direct link to specific mRNAs. (b) It is possible that longer RNAs will have more ac4C and are also more likely to be in SGs, without any causal relationship. This association is at best correlative. (c) Similar to

the comments for Fig. 2, use of the TEs calculated by Arango et al. to gauge translation in the current study is unjustified. In addition to not considering cellular variability between the studies, I am concerned that the authors are treating the results from Arango et al. as though the mapped locations are 100% stoichiometry, which is unlikely to be the case. The authors also have not considered the possibility that the mRNAs identified by Arango et al. are not acetylated in the SGs studied here. In addition, Arango et al. reported that ac4C location in mRNAs can influence TE, which the authors also have not considered. (e) The authors present an interesting correlation between mRNAs that were shown by Arango et al. to be ac4C(+) and SG enrichment. However, in addition to not validating ac4C status, questions remain. For example, Arango et al. reported that NAT10 KO cells proliferated more slowly. This could conceivably alter SG composition and additional controls are required. For example, in addition to validating ac4C status, the authors should examine several SG enriched mRNAs that are ac4C(-).

Figure 4. (a-b) As detailed in the Figure 1 comments, the IF needs much improving. The presented images are not convincing. Why not also stain for ac4C and G3bp1 to directly show colocalization? (c-d) Though it is somewhat intriguing that AHNAK levels in SGs are decreased in NAT10 KO cells, additional controls are required. At a minimum, other ac4C(+) mRNAs that have been confirmed in the authors hands should be examined along with several ac4C(-) controls.

Figure 5. (a) The authors have not provided any data supporting the success of the affinity purification. These results are vital to assessing the validity of the presented results. (b) The relevance of this simulation is unclear. (c) Critical controls are missing, including evidence that ac4C has been successfully titrated in the oligos and western blots for other factors that are not expected to bind ac4C (d) ac4C(-) mRNAs that have been confirmed by the authors should be included as control. (e) Higher resolution images are required, as described above.

In summary, Kudrin and colleagues present an interesting connection between ac4C and SGs. However, the study is underdeveloped in its current form and additional investigation is required to bolster their claims. Supporting QC, additional controls and increased rigor throughout would strengthen the overall significance of the study. At a very minimum, the authors will need to present direct evidence connecting ac4C and SGs. Detailed analysis of a few examples could be pursued to achieve this.

Referee #2:

Kudrin et al report enrichment of acetylated mRNA in SGs in HeLa cells and that this modification shapes the distinct SG transcriptome. The core observation of the study is very interesting and novel, however, further validation work is needed to fully flesh out the dataset. In addition, manuscript requires significant restructuring, in order to clearly convey the message. An important question of the core/shell RNA localization also remains - some of the data presented are difficult to reconcile with what we know about the core/shell SG organization.

Major:

1. All figure legends require major revision - to remove unnecessary narrative ("we used .." and similar) and references - these should be in the main text, which should make the legends more concise and specific.
2. How was ac4C detected in Fig1 - no detail could be found in text, legend, or methods.
3. An alternative approach is required validate the effect of NAT10 LoF on RNA recruitment in SGs, e.g. siRNA mediated knockdown - at least to confirm the core findings of the study (i.e. Fig3-5). Constitutive loss of NAT10 and clonal selection can be associated with defects in various pathways affecting SGs indirectly.
4. Line 51: "RNA can drive ... RNA-binding protein localization to SG" - the evidence for this claim is currently too preliminary, from just one validated candidate.
5. Control for endogenous SG purification using G3BP1 antibody is not indicated (IgG?).
6. Fig1e - intensity of G3BP staining - this readout requires clarification. Was the total G3BP1 fluorescence in cells quantified? If so, it can't be considered a measure of SG assembly. SG formation efficiency should be instead quantified using SG number (with a size threshold(s)) and G3BP1+ foci area as the metrics. Since the study is heavily relying on the premise of normal SG assembly in NAT10 KO cells, comprehensive characterisation of SG formation in this line (and an independent LoF NAT10 model) is warranted.
7. Fig1. The purpose of using NAT10 in determining SG localisation of ac4C in a NAT10 KO cells, where this modification is severely decreased, is not obvious to me. I suggest to restructure the figure keeping the data for ac4C for WT only and providing panels for lack of NAT10 on SGs separately.
8. Fig1. Other stressors and at least one more SG marker are required. Are eIF2a-independent SGs (e.g. Roc A, silvestrol or sorbitol-induced) also accumulate ac4C to the same extent?
9. "Stress granule purification in HeLa cells is highly reproducible" part is essentially an approach validation experiment, and this data should not be presented as a separate subheading but rather merged with the following section, with the figure either provided in Suppl material/extended data, or merged with Fig3.
10. Line 119: "significantly higher fraction than expected" - an appropriate statistical test for enrichment is required.
11. "RNA localization detected with smFISH" subheading - please revise to reflect the findings (e.g. Acetylated mRNA partitioning to SGs is confirmed by smFISH, or similar).
12. Fig4 - a SG marker for this study is essential, to confirm the identity of AHNAK foci (e.g. a polyA RNA probe) and

demonstrate that puromycin equally enhances SG assembly in WT and KO cells (with quantification). Further, the conclusion is made based on a single mRNA, at least one additional RNA is required.

13. Fig5 - WB requires better labelling. MFI - no detail how this was determined, either in the methods or legend. As in comment 6, a different SG quantification approach may be required. G, H, I - graphs need a title. G3BP1 shows diminished fluorescence in KO cells in E - is this a true phenotype?

14. Fig5-relevant section: additional candidates (other than NOP58) should be validated.

15. Fig5A shows depletion of known SG proteins from acetylated RNA binders, e.g. hnRNP A2/B1 and RBM proteins - how can this be reconciled with the authors' model?

16. It is important to address the link between core SG proteins such as G3BP1 and UBAP2L and RNA acetylation. Methods section suggests that non-crosslinked SGs were purified - i.e. SG core but not the shell. Do results of this study imply that acRNA is mainly/only localised in the core? If this is the case, the lack of SG core protein enrichment in the RNA pulldowns is surprising and should be further looked into.

Minor:

1. 37: "their formation, dispersal and function remain largely unclear" - it is difficult to agree with this statement, please refer to multiple recent published work.

2. 116: "the SG RNA content in NAT10 KO cells is less pronounced.." - please revise wording. Also here, [9] - format different from other references.

3. Fig 3b legend - the color code information should be added to the figure and removed from the legend.

4. Sequences of in vitro transcribed RNA oligos are best presented as a table rather than in the methods text.

5. TE (translation efficiency) should be spelled out in Fig3d.

6. "Figure 5. NOP58 localization to SG is dependent on ac4C" legend - should be "NOP58 protein localization is dependent on ac4C modification" or similar

7. 230: something is missing from this sentence

8. Could not find Suppl Table 3 (with oligo sequences)

Referee #3:

Kudrin et al. investigate the role of cytidine acetylation on RNA localization to stress granules (SG). They report that (ac4C) sites are enriched in SG-targeted mRNAs. They further show that the SG protein NOP58 binds ac4C, and its recruitment to SGs is lost in cells. As mentioned by the authors, the mechanisms of RNA partitioning to SG remains elusive. This work thus provides new insight into the role of RNA modifications in modulating the recruitment of RNA and RNA binding proteins to SGs.

Overall, the main statements of the paper are supported by good pieces of evidence. However, as will be detailed below, in several instances the presented data often lack validation elements, experimental details and analyses, or complementary strategies, which render the study incomplete and less convincing. Moreover, the writing could be significantly improved by giving more relevant information in the introduction, results section and figure legends, which are not detailed enough to fully understand how the experiments were carried out. Finally, many of the figure panels in the main figures and supplements are of very poor resolution and not appropriate for publication, thus a thorough reconstruction of the figures needs to be performed.

Major points:

- Throughout the paper, the authors use Nat10 KO cells obtained from another group, but no data is presented validating that these are indeed Nat10 deficient. There would be space in Figure 1 to add appropriate Western gels confirming the absence of Nat10. Also, it seems puzzling that ac4C RNA is still detected in SGs in cells lacking Nat10, which would suggest the involvement of another acetyltransferase in carrying out this modification. The figure would be strengthened by the addition of data profiling the general levels of ac4c-modified RNA in control vs Nat10 KO cells, for instance using dot blots.

- In Figure 2, the authors provided no validation data confirming the efficiency of SG purification. Addition of panels showing a schematic of the purification procedure, as well as purification efficiency, would strengthen the observations. In the volcano plots presented in panels A and B, the authors should consider highlighting the dots corresponding to known mRNAs with ac4C modifications, which would better convey their enrichment behavior in SGs from control cells, and whether these properties are altered in Nat10 KO cells. Regarding data analysis in this figure, it is unclear how the cut-off used to determine SG enriched transcripts was set (lines 100-110). A cut-off of 2 is used for the comparison with previously published data, providing a significant overlap; but then the authors decide to use a cut-off of 4 for the rest of the analysis. Is the new set of enriched transcripts comparable to the published data? This is important since SG purification is not biochemically validated.

- In Figure 3, perhaps the best way to show the correlation between ac4C and SG localization would be to plot the number of ac4C sites vs the logFC (line 124 and figure 3C). Using dot sizes to indicate the number of ac4C makes the current graph difficult to read, while the expression level (FPKM axis) is not relevant in this figure, since it is not mentioned in the text. Also, what fraction of acetylated RNA would be expected in SGs if ac4C did not bias RNA recruitment (line 119)? What is the percentage of acetylated transcripts in the total fraction? The author should comment on why GAPDH mRNA would undergo a

large shift in SG enrichment upon NAT10 depletion while it does not bear any ac4C site (figure 3E and line 150).

- Figure 4 would benefit from the inclusion of a marker to label stress granules in conjunction with the FISH probes, which would allow the authors to more robustly identify SG structures based on labeling with a marker such as G3BP1. In this manner, they could more readily compare the localization behavior of HANAK and GAPDH.

- The current version of Figure 5 suffers from a lack of detail. The authors should include a schematic of how the interactor purifications were conducted. The results section mentions that these assays were conducted with in vitro synthesized RNA bearing different ratios of ac4C vs regular C, but it is unclear how all the data was integrated to general a list of top candidates. The authors should consider presenting the interactomic data in one figure and provide a more detailed schematic of the procedure, how the samples were analyzed, general stats of the data (i.e. what was the total number of proteins detected in each condition and how was the data filtered to general full lists of ac4C interactors, etc). Also, were there any surprises, e.g. proteins that are not expected to interact with RNA, etc? The data on Nop48 could be presented as a separate Figure6.

We thank the reviewers for their constructive suggestions for improving the manuscript. We have incorporated most points in the new manuscript and addressed all points in the following section. In the revised manuscript corrections and addition are typed in red.

Referee #1:

This manuscript by Kudrin and colleagues reports on a potential relationship between stress granules and the presence of ac4C in mRNA. As little is known of how mRNAs are localized to discrete cellular compartments, the questions pursued are of inherent significance. Through a combination of IF, RNA-seq, mass spec and transcript-specific analyses, the authors conclude that mRNAs are targeted to SGs in a NAT10-dependent manner. Though an appealing model in principle, substantial data issues cast doubt on the directness of the shown associations and overall relevance, as described in the below comments.

-> We thank the reviewer for their enthusiasm and specific criticism that has helped make the manuscript even stronger to demonstrate an involvement of ac4C in mRNA stress granule localization. One point that stands out is the discussion of the use of TE data from published work. The reviewer has valid points that these data could ideally be stronger, but is hampered by several aspects. The method used is not quantitative, is not transcript-specific and is not done on single cells to properly understand the stoichiometry and dynamics of ac4C modifications. The method is, however, the state-of-the-art and developing an improved method that meets all the requirements above is beyond the scope of this work. If we were to redo the TE experimental steps and analysis for isolated SG we would encounter the same limitation and it would at best show whether mRNA are acetylated within SG. Our data with mass spec clearly show they are. We have taken the reviewer's concern into consideration as specified under each point and added a comprehensive discussion of these limitations to the text in the context of our conclusions.

Major comments:

Figure 1. (a-d) The provided IF images are of insufficient quality to support the authors' claims.

-> The initial submission contained pdf versions of the images which we have learned gives insufficient quality. We have provided figures in .tif format in the revised manuscript, which give a much higher resolution.

Additional controls are required to ensure that the ac4C staining is specific, including an RNase(+) condition. In addition, quantification of % ac4C overlap in SGs as compared to background should be examined.

-> We have included an RNase control (Figure S1E) as well as quantification of ac4C enrichment (Figure 1d) and ac4C-enriched transcripts in SG (Figure 4d-e and Figure S3e-f) with a better description of how it was done.

As presented, it appears nucleoli are disrupted in response to sodium arsenite, which could create the perception of greater SG overlap. Much higher resolution pictures should be provided to better demonstrate co-localization.

-> Again, we apologize for the insufficient image quality in the initial submission. We have provided tif pictures in the revised manuscript that provide a higher resolution and hope that the presented data are of sufficient quality to evaluate the integrity of the nucleoli.

(f) It is difficult to assess the relevance of these results in the absence of any data showing the efficacy of SG RNA enrichment. It is also unclear what the plot is showing. Are these mass spec results? If so, how are the enrichment values calculated and normalized?

-> We thank the reviewer for pointing this out and have clarified the plots better in the revised text. The purification of SG RNA is done according to Khong et al. ¹ and validated by smFISH of RNA seq data as recommended in the protocol. In addition, RNA seq is compared to other published studies as described in figure 2c-f, to show reproducibility of methods between labs and cell lines, showing a reproducible enrichment of core SG mRNAs in our experimental work, confirming efficient SG purification. We have added a more specific description of the mass spec analysis of the samples.

Supplementary Fig. 1. The relative enrichment of ac4C in RNA purified from SGs is intriguing, but it is unclear how the mass spec is normalized. Additional information including controls should be included to ensure sample purity.

-> We apologize for the confusion and have added a more specific description of the mass spec analysis of the samples in the revised manuscript. The quantification of m6A and m7G are used as controls for sample purity as they have different abundances on different possible contaminating RNA species as well as being very common on mRNA.

Figure 2. The overall relevance of this figure is unclear. (c-d) I don't follow the rationale for examining SG RNAs in different cells. The presented results suggest that the transcripts that are found in SGs vary substantially between cell types, which agrees with my understanding.

-> We thank the reviewer for assessing these data, that we find of relevance for assessing the quality and reproducibility of the purification of SG. It is central to examining the function of SG and comparing to our data to other published studies using the same method for understanding the robustness of the protocol and the confidence level of the transcripts we identify as SG enriched. We have used the comparison of enriched core transcripts from Khong et al., 2017 ² as a benchmark for SG purification and to establish a confidence level for further analysis. We have added a more concise description of the relevance of Figure 2 and the comparison between SG transcriptomes from different studies to the revised manuscript.

(e) It has been reported by others that SGs are transient and that their composition can vary substantially in different conditions. With this in mind, the use of TE results that were performed by another lab at another time does not seem appropriate. In addition, ac4C is believed to be a low stoichiometry modification and there is no way of knowing whether the TEs that were measured in bulk relate to the modified pool of mRNA. Relatedly, as the authors have not directly mapped ac4C in SG mRNA, they cannot be confident whether the mRNAs they are examining are in fact acetylated.

-> We thank the reviewer for this very relevant point. We have used TE data from bulk mRNA using the same cells as for our study. The reviewer has valid points that these data could ideally be stronger, but this aspect is hampered by several factors. As the reviewer states there is no way of knowing the stoichiometry of acetylation on each transcript, as the method used is not quantitative, is not transcript-specific and is not done on single. The method is, however, the state-of-the-art and developing an improved method that meets all the requirements above is beyond the scope of this work. If we were to redo the TE experimental steps and analysis for isolated SG we would encounter the same limitations and it would at best show whether mRNA are acetylated within SG but again not provide data on stoichiometry or variations of ac4C sites between transcripts.

Figure 3. (b) As the authors did not map ac4C in SGs, they cannot reasonably conclude that the mRNAs that they detect in SGs are in fact ac4C modified. Though the mass spec shown in

Supplemental Figure 1 suggests that some SG resident RNAs are acetylated, the presented data do not establish a direct link to specific mRNAs.

-> We appreciate the reviewer's effort to clarify this important point in the manuscript. We show in our data that there is a correlation between SG localization and ac4C levels in bulk mRNA. It is reasonable to assume that possible ac4C modification sites do not change between cytoplasm and SG. The reviewer is right that these data do not provide a direct link to specific mRNAs, which is why we have included the validation experiments with smFISH, providing a direct link between ac4C and localization of specific mRNAs. We also point the attention to the fact that the ac4C mapping of Arango et al., 2022³ is on bulk mRNA and does not take variations over transcripts into consideration, as stated in the introductory and previous comments to the reviewer. Hence, even with comparative data using state-of-the-art techniques between cytoplasm and SG we could not for certain say that the transcripts were acetylated the same way and with the same stoichiometry. We have taken care in the revised text not to state that the transcripts are acetylated exactly the same, nor necessarily with 100% stoichiometry, but emphasize that the transcripts contain validated ac4C sites that are likely to be acetylated, and these transcripts correlate with SG localization.

(b) It is possible that longer RNAs will have more ac4C and are also more likely to be in SGs, without any causal relationship. This association is at best correlative.

-> The reviewer has a very valid point that is difficult to address directly. As several studies have reported length as a determining factor for SG localization, our work is the first to examine the role of ac4C modifications of mRNA. Inspired by a recent paper on SG localization of mRNA and the impact of m6A⁴ we plotted cumulative frequency curves of number of ac4C sites and mRNA length, respectively, as a function of SG enrichment for both WT and NAT10 KO cells (Figure 3d-g). Here, we see a clear effect of ac4C sites on SG recruitment that is decreased in NAT10 KO cells. Of importance, we see that the localization of long mRNAs to SG is abrogated when ac4C levels are decreased in the NAT10 KO cells, proposing, in line with the arguments of⁴ that ac4C has a direct effect on increasing localization of long mRNAs to SG.

(c) Similar to the comments for Fig. 2, use of the TEs calculated by Arango et al. to gauge translation in the current study is unjustified. In addition to not considering cellular variability between the studies, I am concerned that the authors are treating the results from Arango et al. as though the mapped locations are 100% stoichiometry, which is unlikely to be the case. The authors also have not considered the possibility that the mRNAs identified by Arango et al. are not acetylated in the SGs studied here. In addition, Arango et al. reported that ac4C location in mRNAs can influence TE, which the authors also have not considered.

-> We thank the reviewer for pointing this crucial aspect out. As also stated in previous replies and in the initial comment to the reviewer, the ac4C mapping method, although state-of-the-art, has several limitations that makes the proposed analysis unfeasible. We show in our data that there is a correlation between SG localization and ac4C levels in bulk mRNA. It is reasonable to assume that possible ac4C modification sites do not change between cytoplasm and SG. In addition, we do not claim that there is 100% stoichiometry, but state that the mRNAs enriched in SG tend to have more (possible) ac4C sites. We have emphasized this point in the results and discussion sections to avoid misunderstandings. Also, including the puromycin conditions for SG induction we consider the possible effect of ac4C on TE and SG formation, show that acetylated mRNAs have a distinct TE profile and have added additional text on this in the discussion. We have discussed at line 298-301 in the revised manuscript the reported effect of ac4C on translation mentioned by the reviewer.

(e) The authors present an interesting correlation between mRNAs that were shown by Arango et al. to be ac4C(+) and SG enrichment. However, in addition to not validating ac4C status, questions remain. For example, Arango et al. reported that NAT10 KO cells proliferated more slowly. This could conceivably alter SG composition and additional controls are required. For example, in addition to validating ac4C status, the authors should examine several SG enriched mRNAs that are ac4C(-).

-> The reviewer has a fair point that the NAT10 KO cells are different in more aspects than simply having less ac4C. To include this in the analysis we have focused on transcripts that are enriched in WT cells, and downplayed the comparison between WT and NAT10 KO cells to demonstrate a difference in SG formation and mRNA localization on a high level. In our smFISH data we have included GAPDH as a non-ac4C transcript. Due to the cost of smFISH probes and comments from the other reviewers we have focused on showing another ac4C+ transcript with smFISH in the revised manuscript, rather than including several non ac4C transcripts.

Figure. 4. (a-b) As detailed in the Figure 1 comments, the IF needs much improving. The presented images are not convincing. Why not also stain for ac4C and G3bp1 to directly show colocalization? (c-d) Though it is somewhat intriguing that AHNAK levels in SGs are decreased in NAT10 KO cells, additional controls are required. At a minimum, other ac4C(+) mRNAs that have been confirmed in the authors hands should be examined along with several ac4C(-) controls.

-> We thank the reviewer for this critical assessment. We have included co-staining for G3BP1 as well as PABP with ac4C staining in the revised manuscript. In the light of price for the smFISH probes we have focused on adding another ac4C+ transcript in the revised manuscript to demonstrate changes in localization for specific transcripts.

Figure 5. (a) The authors have not provided any data supporting the success of the affinity purification. These results are vital to assessing the validity of the presented results.

-> We thank the reviewer for this comment. In the manuscript we have included validation of the identified candidate, NOP58, by WB in independent experiments in Figure 5b, and added a GAPDH control for specificity. Following discussions with the editor we have decided to downplay the data on protein interactions and focus on the RNA part of ac4C in SG localization. We have restructured Figure 5 slightly and modified the discussion to reflect on the question of protein localization to SG dependent of ac4C RNA modifications.

(b) The relevance of this simulation is unclear.

-> We thank the reviewer for pointing this out and have elaborated the description in the revised manuscript. In brief, we see that mRNAs with higher number of ac4C sites are predicted to bind NOP58 better, supporting a biologically relevant binding between NOP58 and mRNAs with possible ac4C sites. The correlation would be more biologically relevant if we could include the impact of ac4C on the interaction propensity, but this feature is to the best of our knowledge not yet available for any tool. The relevance is therefore to add a theoretical support to the experimental data on NOP58 binding to ac4C modified RNAs.

(c) Critical controls are missing, including evidence that ac4C has been successfully titrated in the oligos and western blots for other factors that are not expected to bind ac4C

-> We thank the reviewer for noticing this, and have added quantification by mass spec of ac4C incorporation into the oligos in Figure S4c-d to the revised manuscript, and added WB for GAPDH as a protein not binding to ac4C in our pull-down in Figure 5b.

(d) ac4C(-) mRNAs that have been confirmed by the authors should be included as control.

-> In the revised manuscript we have added the ac4C modified transcript MKI67 and included GAPDH as non-acetylated negative control.

(e) Higher resolution images are required, as described above.

-> Throughout the revised manuscript we have updated with new pictures and pictures with higher resolution.

In summary, Kudrin and colleagues present an interesting connection between ac4C and SGs. However, the study is underdeveloped in its current form and additional investigation is required to bolster their claims. Supporting QC, additional controls and increased rigor throughout would strengthen the overall significance of the study. At a very minimum, the authors will need to present direct evidence connecting ac4C and SGs. Detailed analysis of a few examples could be pursued to achieve this.

Referee #2:

Kudrin et al report enrichment of acetylated mRNA in SGs in HeLa cells and that this modification shapes the distinct SG transcriptome. The core observation of the study is very interesting and novel, however, further validation work is needed to fully flesh out the dataset. In addition, manuscript requires significant restructuring, in order to clearly convey the message. An important question of the core/shell RNA localization also remains - some of the data presented are difficult to reconcile with what we know about the core/shell SG organization.

-> We thank the reviewer for their interest in our work. We have addressed the points below in a point-by-point response. A central point is the NAT10 cell line and how good a cell line it is for the study. Using siRNAs against NAT10 other studies have shown that ac4C levels are higher (30-50% of WT), and that cell viability is severely affected⁵, which we also observe in our work to address this question, where knock-down efficiency in HeLa cells was in the range of 30 per cent while being efficient in other cell types. One explanation could be that the NAT10 KO cell line does not deplete NAT10 completely but introduces a mutation in exon 5 that might maintain other essential functions of NAT10. With the remaining ac4C levels (20 per cent) in the NAT10 KO cell line, and the viability issues using siRNA, we would assume less biologically relevant data and a less complete depletion using siRNA, complicating the interpretation of the data. We have therefore not included the proposed experiments with siRNA in the revised manuscript, but have added a further discussion on the generation of the NAT10 KO cell line, also supported by RNA mass spec (Figure S1b) and WB of the NAT10 protein (Figure S1c).

Major:

1. All figure legends require major revision - to remove unnecessary narrative ("we used .." and similar) and references - these should be in the main text, which should make the legends more concise and specific.

-> We thank the reviewer for their attention to detail and have thoroughly revised all figure legend.

2. How was ac4C detected in Fig1 - no detail could be found in text, legend, or methods.

-> We apologize for this omission in the initial submission. In the revised manuscript, we detail how the RNA modifications in Figure 1d and Figure S1f-j were detected by mass spec. In Figure 1a-b ac4C is detected by immunostaining as detailed in the manuscript. The details of the experiments have been added to the revised manuscript and in the methods section.

3. An alternative approach is required to validate the effect of NAT10 LoF on RNA recruitment in SGs, e.g. siRNA mediated knockdown - at least to confirm the core findings of the study (i.e. Fig3-5). Constitutive loss of NAT10 and clonal selection can be associated with defects in various pathways affecting SGs indirectly.

-> We thank the reviewer for this comment and have attempted siRNA mediated knock-down of NAT10 with limited efficiency and significant impact on cell viability, as also detailed in the opening comment to the reviewer. We therefore cannot consider experiments using siRNA against NAT10 as more physiological relevant than the NAT10 KO cell line. In addition, current literature suggests that the impact on ac4C levels following knock-down of NAT10 with siRNA is limited in HeLa cells and confirms unwanted side-effects on cell proliferation and apoptosis following NAT10 knock-down by siRNA⁵. We have added a better description of how the NAT10 KO cell line is generated to explain why this difference exists, as it is an induced mutation in exon 5 of NAT10 rather than a complete depletion. We have also added determination of ac4C levels and western blot of NAT10 to the

revised manuscript (Figures S1b-c). In our work we demonstrate that ac4C modified transcripts are localized to SG in WT HeLa cells and provide a suite of analyses to support this finding. The NAT10 KO cell line is used for comparison and to bolster the conclusions but are as such not a prerequisite for the conclusions of the study. We therefore recognize the reviewer's suggestion but find that the manuscript as presented in the revised version offers strong enough support for our findings, also in the light of complications with depleting NAT10 with siRNA in a physiologically meaningful HeLa cell line model.

4. Line 51: "RNA can drive ... RNA-binding protein localization to SG" - the evidence for this claim is currently too preliminary, from just one validated candidate.

-> We recognize that this section should not have such a central place in the manuscript, and have rephrased the statement to focus on the role of ac4C on mRNA localization. In the revised manuscript we have put less emphasis on the identified ac4C interacting proteins, and moved the large-scale analysis to Figure S4.

5. Control for endogenous SG purification using G3BP1 antibody is not indicated (IgG?).

-> For purification of SG we have followed the protocol by Khong and Parker 1 with the validation by smFISH of selected mRNAs. In addition, we have added the comparison to other published studies in Figure 2c-f to show the reproducibility and validity of our SG purification, as well as determining the confidence level of identified mRNAs partitioned to SG.

6. Fig1e - intensity of G3BP staining - this readout requires clarification. Was the total G3BP1 fluorescence in cells quantified? If so, it can't be considered a measure of SG assembly. SG formation efficiency should be instead quantified using SG number (with a size threshold(s)) and G3BP1+ foci area as the metrics. Since the study is heavily relying on the premise of normal SG assembly in NAT10 KO cells, comprehensive characterisation of SG formation in this line (and an independent LoF NAT10 model) is warranted.

-> We thank the reviewer for stressing this. The quantification has been done the way the reviewer suggests with size threshold and quantification of the G3BP1 foci. We have expanded the experimental details on quantification of immunofluorescence in the methods section to elaborate on this.

7. Fig1. The purpose of using NAT10 in determining SG localisation of ac4C in a NAT10 KO cells, where this modification is severely decreased, is not obvious to me. I suggest to restructure the figure keeping the data for ac4C for WT only and providing panels for lack of NAT10 on SGs separately.

-> We thank the reviewer for this comment and have reconsidered the main and extended view Figures for these data. We agree that WT cells are most central to the conclusions drawn in this paper and have moved the NAT10 KO staining from the initial Figure 1 to Figure S1d.

8. Fig1. Other stressors and at least one more SG marker are required. Are eIF2a-independent SGs (e.g. Roc A, silvestrol or sorbitol-induced) also accumulate ac4C to the same extent?

-> We have included co-staining with both SG markers G3BP1 and PABP upon arsenite and sorbitol-induced SG formation in Figure S1a. While we see co-localization of ac4C and G3BP1 and PABP following sorbitol stress, partitioning of ac4C mRNA to SG seems less complete with sorbitol, which we have added to the results section.

9. "Stress granule purification in HeLa cells is highly reproducible" part is essentially an approach validation experiment, and this data should not be presented as a separate subheading but rather merged with the following section, with the figure either provided in Suppl material/extended data, or merged with Fig3.

-> We thank the reviewer for this suggestion to restructure the manuscript. The data are relevant to address comments on purification efficiency and reproducibility, but we agree that it should not have its own subheading. We have restructured the text to give it a less central position.

10. Line 119: "significantly higher fraction than expected" - an appropriate statistical test for enrichment is required.

-> We have included that the statistical test is a one-way ANOVA.

11. "RNA localization detected with smFISH" subheading - please revise to reflect the findings (e.g. Acetylated mRNA partitioning to SGs is confirmed by smFISH, or similar).

-> The subheading has been changed.

12. Fig4 - a SG marker for this study is essential, to confirm the identity of AHNAK foci (e.g. a polyA RNA probe) and demonstrate that puromycin equally enhances SG assembly in WT and KO cells (with quantification). Further, the conclusion is made based on a single mRNA, at least one additional RNA is required.

-> In Figure S3a we have added staining with a polyA RNA probe and overlaid it with the SG found by G3BP staining. In Figure S1a we have in addition added SG marker PABP to increase confidence in the observed SG. In Figure 4a-c we show the impact on mRNA localization to SG by arsenite, and quantify this in Figures 4d-e and S3e-f, where we see that puromycin affects SG to a comparable degree in WT and NAT10 cells, albeit with lower mRNA localization in NAT10 KO cells. We have also added smFISH for the highly acetylated MKI67 transcript to support the conclusion.

13. Fig5 - WB requires better labelling. MFI - no detail how this was determined, either in the methods or legend. As in comment 6, a different SG quantification approach may be required. G, H, I - graphs need a title. G3BP1 shows diminished fluorescence in KO cells in E - is this a true phenotype?

-> We have labeled the WB better and added a GAPDH control for a non-binding protein. We have added details on MFI in the methods. Regarding quantification of SG, panel C shows pull-down of transcripts with NOP58 compared to IgG and does not include SG purification. Labels have been added for panels F-H (new Figure 5). The IF in panel D for NAT10 KO does not have reduced fluorescence, which is also not the case for Figure 1, meaning that it is not a true phenotype but experimental variation.

14. Fig5-relevant section: additional candidates (other than NOP58) should be validated.

-> This is a very valid point that we have consulted with the editor. We find it outside the scope of this manuscript to validate and characterize further ac4C binding proteins. We have rephrased the statement (Reviewer comment 4) to put less emphasis on the protein relocalization and keep a focus on ac4C modified mRNA. We have also moved initial Figure 5A to revised Supplementary Table 2, to put more emphasis on the RNA aspects of ac4C-mediated partitioning to SG, that we would like to be the main message of the manuscript.

15. Fig5A shows depletion of known SG proteins from acetylated RNA binders, e.g. hnRNP A2/B1 and RBM proteins - how can this be reconciled with the authors' model?

-> We thank the reviewer for this insightful comment. The data are from proteins purified from whole-cell extract using an ac4C-modified oligo. The explanation why we see a depletion of these known SG proteins are that ac4C has other functions than to mediate SG formation, and that not all SG proteins bind directly to ac4C. In the revised manuscript we have moved Figure 5a to Supplementary Table 2 and focus on NOP58 in the main text, to give the protein interactions a less central role and focus on RNA acetylation and its role in localization to SG.

16. It is important to address the link between core SG proteins such as G3BP1 and UBAP2L and RNA acetylation. Methods section suggests that non-crosslinked SGs were purified - i.e. SG core but not the shell. Do results of this study imply that acRNA is mainly/only localised in the core? If this is the case, the lack of SG core protein enrichment in the RNA pulldowns is surprising and should be further looked into.

-> The reviewer is right that our study addresses the SG core by using non-crosslinked SG. That does not imply that ac4C is only localized in the core, as our data do not address the SG shell. mRNA with ac4C can be localized to several subcellular compartments and we do not attempt to claim that they are specific to SG cores, rather we conclude that ac4C on mRNA promotes their localization to SG cores, which does not preclude localization to the shell or other cellular compartments.

Minor:

1. 37: "their formation, dispersal and function remain largely unclear" - it is difficult to agree with this statement, please refer to multiple recent published work.

-> We have rephrased this sentence, with focus on RNA modifications. We have also added several references to work on m6A in SG localization of RNA.

2. 116: "the SG RNA content in NAT10 KO cells is less pronounced.." - please revise wording. Also here, [9] - format different from other references.

-> We have rephrased this sentence and corrected the format of the reference.

3. Fig 3b legend - the color code information should be added to the figure and removed from the legend.

-> We have added the legend to the figure and corrected the legend.

4. Sequences of in vitro transcribed RNA oligos are best presented as a table rather than in the methods text.

-> We have included a table with the sequences of in vitro transcribed RNA oligos.

5. TE (translation efficiency) should be spelled out in Fig3d.

-> We have corrected the label in Figure 3d (now Figure 3i).

6. "Figure 5. NOP58 localization to SG is dependent on ac4C" legend - should be "NOP58 protein localization is dependent on ac4C modification" or similar

-> We have corrected the legend.

7. 230: something is missing from this sentence

-> We have thoroughly revised the manuscript for typos and missing words.

8. Could not find Suppl Table 3 (with oligo sequences)

-> Supplementary Table 3 should have been included in the initial submission. We have made sure it has been included in an updated version in the revised manuscript.

Referee #3:

Kudrin et al. investigate the role of cytidine acetylation on RNA localization to stress granules (SG). They report that (ac4C) sites are enriched in SG-targeted mRNAs. They further show that the SG protein NOP58 binds ac4C, and its recruitment to SGs is lost in cells. As mentioned by the authors, the mechanisms of RNA partitioning to SG remains elusive. This work thus provides new insight into the role of RNA modifications in modulating the recruitment of RNA and RNA binding proteins to SGs.

Overall, the main statements of the paper are supported by good pieces of evidence. However, as will be detailed below, in several instances the presented data often lack validation elements, experimental details and analyses, or complementary strategies, which render the study incomplete and less convincing. Moreover, the writing could be significantly improved by giving more relevant information in the introduction, results section and figure legends, which are not detailed enough to fully understand how the experiments were carried out. Finally, many of the figure panels in the main figures and supplements are of very poor resolution and not appropriate for publication, thus a thorough reconstruction of the figures needs to be performed.

-> We thank the reviewer for their enthusiastic comments on our manuscript. We have revised the points raised and improved writing, organization and picture resolution for all figures.

Major points:

- Throughout the paper, the authors use Nat10 KO cells obtained from another group, but no data is presented validating that these are indeed Nat10 deficient. There would be space in Figure 1 to add appropriate Western gels confirming the absence of Nat10. Also, it seems puzzling that ac4C RNA is still detected in SGs in cells lacking Nat10, which would suggest the involvement of another acetyltransferase in carrying out this modification. The figure would be strengthened by the addition of data profiling the general levels of ac4c-modified RNA in control vs Nat10 KO cells, for instance using dot blots.

-> We have included a WB of NAT10 in WT and NAT10 KO cells showing the depletion levels (Figure S1c), and detailed that the cell line is made by mutations in exon 5 of the NAT10 gene, thus expressing a certain level of non-functional NAT10. We have also included quantification of ac4C in WT and NAT10 KO cells by mass spec in mRNA (Figure S1b). It has previously also been shown that NAT10 KO decreases ac4C levels with 80%⁶, suggesting as the reviewer proposes that another acetyltransferase exists, the identity of which is unknown. We have added more description of NAT10 as acetyltransferase and the generation of the KO model to the introduction.

- In Figure 2, the authors provided no validation data confirming the efficiency of SG purification.

-> For purification of SG we have followed the protocol by Khong and Parker¹ with the validation by smFISH of selected mRNAs. In addition, we have added the comparison to other published studies in Figure 2c-f to show the reproducibility and validity of our SG purification, as well as determining the confidence level of identified mRNAs partitioned to SG.

Addition of panels showing a schematic of the purification procedure, as well as purification efficiency, would strengthen the observations.

-> We have added a schematic showing the purification procedure (Figure S2a), and detailed the assessment of the quality of purifications in the revised manuscript.

In the volcano plots presented in panels A and B, the authors should consider highlighting the dots corresponding to known mRNAs with ac4C modifications, which would better convey their enrichment behavior in SGs from control cells, and whether these properties are altered in Nat10 KO cells. Regarding data analysis in this figure, it is unclear how the cut-off used to determine SG enriched transcripts was set (lines 100-110). A cut-off of 2 is used for the comparison with previously published data, providing a significant overlap; but then the authors decide to use a cut-off of 4 for the rest of the analysis. Is the new set of enriched transcripts comparable to the published data? This is important since SG purification is not biochemically validated.

-> We thank the reviewer for this suggestion. We have added indication of some of the known ac4C modified mRNAs used in this study in Figure 2a-b to show their relative enrichments in both WT and NAT10 KO cells. We have detailed the choice of different cut-offs better in the manuscript as also stated in Figure 2e-f where we have defined a high-confidence set based on the comparison with published SG purification studies. This means that with the higher cut-off the data are more comparable to other published data on the SG transcriptome.

- In Figure 3, perhaps the best way to show the correlation between ac4C and SG localization would be to plot the number of ac4C sites vs the logFC (line 124 and figure 3C). Using dot sizes to indicate the number of ac4C makes the current graph difficult to read, while the expression level (FPKM axis) is not relevant in this figure, since it is not mentioned in the text. Also, what fraction of acetylated RNA would be expected in SGs if ac4C did not bias RNA recruitment (line 119)? What is the percentage of acetylated transcripts in the total fraction? The author should comment on why GAPDH mRNA would undergo a large shift in SG enrichment upon NAT10 depletion while it does not bear any ac4C site (figure 3E and line 150).

-> We thank the reviewer for pointing this out and have added a better description and four new panels to Figure 3. For the panel c that we have maintained in its original form, FPKM is relevant as expression level of the mRNA could affect partitioning of mRNA into SG, which has been added to the text. The new panels d-g show ac4C sites and mRNA length, respectively, and SG enrichment in cumulative enrichment curves, demonstrating the relevance of ac4C in localization of mRNA to SG, also in the case of long transcripts, substantiating the impact of ac4C on mRNA recruitment to SG.

As shown in Figure S2b there is a high enrichment of ac4C transcripts in SG that increases with number of potential ac4C sites. If ac4C did not bias recruitment this trend would not be observed. While the current methods to map ac4C are not quantitative nor transcript-specific we cannot put a number on the percentage of acetylated transcripts in SG. We have, however, added statistical significance to the overlap between acetylated mRNA and transcripts enriched in SG in our study, as well as increase over expected overlap (Figure 3a). GAPDH shows a shift in the SG enrichment but remains enriched in NAT10 KO cells. The figure in the initial submission showed a too large shift that has been corrected in the revised manuscript. In addition, we generally see lower fold-enrichments of mRNA from NAT10 KO cells, which should also be taken into consideration.

- Figure 4 would benefit from the inclusion of a marker to label stress granules in conjunction with the FISH probes, which would allow the authors to more robustly identify SG structures based on

labeling with a marker such as G3BP1. In this manner, they could more readily compare the localization behavior of HANAK and GAPDH.

-> We thank the reviewer for pointing this out, and have included co-staining of G3BP1 and ac4C throughout the revised manuscript. In addition, we have validated SG with PABP staining and poly(A) smFISH probes (Figures S1a and S3a, respectively).

- The current version of Figure 5 suffers from a lack of detail. The authors should include a schematic of how the interactor purifications were conducted.

-> We have included a schematic of the purification procedure as Figure S4a.

The results section mentions that these assays were conducted with in vitro synthesized RNA bearing different ratios of ac4C vs regular C, but it is unclear how all the data was integrated to general a list of top candidates. The authors should consider presenting the interactomic data in one figure and provide a more detailed schematic of the procedure, how the samples were analyzed, general stats of the data (i.e. what was the total number of proteins detected in each condition and how was the data filtered to general full lists of ac4C interactors, etc). Also, were there any surprises, e.g. proteins that are not expected to interact with RNA, etc? The data on Nop48 could be presented as a separate Figure6.

-> We thank the reviewer for their interest in the protein interaction data. During review and in consultation with the editor we have decided to put less emphasis on the identification of interacting proteins. We have therefore revised the figure and included more detail on the protein purification method and data in Figure S4 as well as in Supplementary Table 2. Of interesting observation we can mention, as commented by reviewer 1, that depletion of known SG proteins e.g. hnRNP A2/B1 and RBM proteins is observed, showing that ac4C is not the only factor important for SG assembly. To give protein interactions a less central role we decided to move this analysis to supplementary data and focus on RNA localization and the one well-characterized interacting protein, NOP58, in the main text and Figure 5.

References

1. Khong, A., Jain, S., Matheny, T., Wheeler, J. R. & Parker, R. Isolation of mammalian stress granule cores for RNA-Seq analysis. *Methods* **137**, 49–54 (2018).
2. Khong, A. *et al.* The Stress Granule Transcriptome Reveals Principles of mRNA Accumulation in Stress Granules. *Molecular cell* **68**, 808-820.e5 (2017).
3. Arango, D. *et al.* Direct epitranscriptomic regulation of mammalian translation initiation through N4-acetylcytidine. *Molecular cell* (2022) doi:10.1016/j.molcel.2022.05.016.
4. Ries, R. J., Pickering, B. F., Poh, H. X., Namkoong, S. & Jaffrey, S. R. m6A governs length-dependent enrichment of mRNAs in stress granules. *Nat Struct Mol Biol* **30**, 1525–1535 (2023).
5. Dalhat, M. H. *et al.* NAT10: An RNA cytidine transferase regulates fatty acid metabolism in cancer cells. *Clin Transl Med* **12**, e1045 (2022).
6. Arango, D. *et al.* Acetylation of Cytidine in mRNA Promotes Translation Efficiency. *Cell* (2018) doi:10.1016/j.cell.2018.10.030.

Dear Ulf,

Thank you for your patience while your ms was re-reviewed at EMBO reports. We have now received the enclosed reports from referees 1 and 2. Unfortunately, referee 3 has not yet sent a report and is unresponsive to our reminders. I am therefore making a decision on your ms now based on the 2 reports we have in order to save you from further loss of time.

I am happy to say that both referees support the publication of your revised study. They only have a few more minor suggestions that I would like you to incorporate before we can proceed with the official acceptance of your manuscript.

A few editorial requests will also need to be addressed:

- Your ms has 5 main figures but is laid out as a full article. Please either combine the results and discussion sections or add one more main figure. You can find more info about our article types in our guide to authors online.
- Please add up to 5 keywords to the ms file.
- Please rename the conflict of interest subheading to "Disclosure Statement and Competing Interests"
- Please remove the author credits from the ms file. All credits need to be entered during ms submission now.
- Please correct the reference format to the EMBO reports style. It needs to be alphabetical and DOIs should only be used for preprints and datasets that have not been published yet.
- Figure 5 looks more like landscape than portrait format, it would be better if you could move panels F-H to below the other panels.
- Fig S2A is called out but there is no such figure - please correct.
- Tables EV1-EV3 are Datasets and need to be uploaded and renamed as such (Dataset EV1, etc); their legends should be removed from the ms file and provided in each Excel file as a separate sheet/tab; the callouts in the ms also need to be corrected.
- This new panel in the source data (SD) checklist needs to be corrected: S1F - are you referring to Figure EV1F?
- Please rename Summary to Abstract
- The manuscript sections should be in the following order: Title page - Abstract & Keywords - Introduction - Results - Discussion - Materials & Methods - Data Availability - Acknowledgments - Disclosure Statement & Competing Interests - References - Figure Legends - Tables with legends - Expanded View Figure Legends.
- Please address the following comments by our data editors on the figure legends:
 1. Please indicate the statistical test used for data analysis in the legends of figures 2a-b; 3h.
 2. Please note that in figures 4e; EV 3e-f; there is a mismatch between the annotated p values in the figure legend and the annotated p values in the figure file that should be corrected.
 3. Please note that the scale bar needs to be defined for figures EV 3c-d.

I would like to suggest some minor changes to the abstract. Please let me know whether you agree with the following:

Stress granules are an integral part of the stress response that are formed from non-translating mRNAs aggregated with proteins. While much is known about stress granules, the factors that drive their mRNA localization are incompletely described. Modification of mRNA can alter the properties of the nucleobases and affect processes such as translation, splicing and localization of individual transcripts. Here, we show that the RNA modification N4-acetylcytidine (ac4C) on mRNA associates with transcripts enriched in stress granules and that stress granule localized transcripts with ac4C are specifically translationally regulated. We also show that ac4C on mRNA can mediate co-localization of the protein NOP58 to stress granules. Our results suggest that acetylation of mRNA regulates localization of both stress-sensitive transcripts and RNA-binding proteins to stress granules and adds to our understanding of the molecular mechanisms responsible for stress granule formation.

EMBO press papers are accompanied online by A) a short (1-2 sentences) summary of the findings and their significance, B) 2-3 bullet points highlighting key results and C) a synopsis image that is exactly 550 pixels wide and 200-600 pixels high (the height is variable). You can either show a model or key data in the synopsis image. Please note that text needs to be readable at the final size. Please send us this information along with the final manuscript.

Referee #1:

This revised manuscript from Kudrin and colleagues is substantially improved and all my major concerns have been adequately addressed. The authors clearly demonstrate the enrichment of acetylated RNAs in SGs and the results represent a significant advance in the fields of RNA modifications and translational regulation. The association between ac4C and SG is intriguing and will certainly spark additional investigations. The revised manuscript is also improved in its general readability. However, I recommend a couple additional clarifications to make the manuscript more accessible to its readers, as outlined below. In addition, Table EV2 is insufficiently annotated. With these changes, I recommend this manuscript for publication.

Line 69-70. It is a stretch to propose that an additional RNA acetyltransferase exist based on the current state of the literature. Certainly, NAT10 depletion results in a profound decrease in ac4C levels at all tested RNA substrates. This is inconsistent with functionally redundant factors. Notably, Arango et al., 2018 showed that residual NAT10 is retained in CRISPR Cas9 deleted HeLa due to splicing around the targeted exon. It is possible that the low level of acetylation in the NAT10 deleted cells stems from the mis spliced protein. I recommend that the authors reword this section to also offer this alternative explanation.

Fig. EV1e. I'm confused by the retention of ac4C signal in the nucleus after treatment with RNase. Is this because the nuclear RNA is inaccessible, or because the antibody is non-specific? I recommend that the authors offer a potential explanation.

Table EV2. It is really not clear what is being presented in this table. What do the various columns mean? An additional tab with sample information should be provided. Were all the RNAs combined and used as bait? Was unmodified RNA used as a control? What control is used for non-specific binding? This table requires additional annotation prior to publication.

Referee #2:

I believe the authors addressed the vast majority of my comments however polishing of text is required, as stated in my original report, the manuscript can be made more concise and readability can be improved, some suggestions are below.

Fig1A: it is good to add RNAs but they can be barely read in purple, please revise colour scheme for this panel.

A bar with "NaAsO2 treatment" label in it was added to many figures. However, this is confusing - this kind of bar would normally indicate an increased concentration, whereas the concentration is constant but the time increases.

Running title: too bold, this modification is not sufficient to localise mRNA to SGs, it has a modulatory effect.

Abstract: "In addition, we show that ac4C on mRNA can mediate co-localization of the protein NOP58 to stress granules." - localisation rather than co-localisation I believe.

Lines 67-70 and 75-79 are essentially repetition. I suggest getting rid of the first mention and simply state that "In human, ac4C is deposited primarily by the acetyltransferase NAT10 (N-acetyltransferase 10) although alternative acetyltransferase(s) for RNA likely also exist." And keep lines 75-79.

Manuscript still requires proofreading. E.g. line 83 "with both the SG markers G3BP and PABP" should be "for both SG markers G3BP and PABP" etc.

Line 89 - indispensable replace with essential

G3BP or G3BP1 - needs consistency. Likewise, SG and SGs.

Line 94 - "in upon arsenite stress." - remove 'in'

Section 2 and 3 'Isolation of SG associated transcripts from WT and NAT10 depleted cells' and 'The stress granule transcriptome

is acetylated' should be merged and section 3 subheading kept.

Line 145 - 'transcripts'

Line 149 - I suggest deleting 'with less pronounced SG transcriptome', it sounds confusing.

Line 232: this sentence sounds awkward, please rephrase.

Line 268: NOP58 protein is binding to acetylated mRNA and is localized to SG - this section should be merged with the previous one, it does not require a separate subheading.

Figure 3 legend is still too long and contains too much non-essential data description, please condense.

Figure 4 legend - please revise to condense, e.g. "smFISH, coupled to G3BP-IF was used to visualise the core transcripts AHNAK and MKI67 and the control transcript GAPDH in SGs (a, b and c, respectively). Red is smFISH probe, green is G3BP and blue is DAPI. Scale bar, 10 μ m. Quantification of the fraction of AHNAK (d) and MKI67 (e) mRNAs in SGs per cytoplasm is also shown. 20 cells per each condition were counted (circles). Data is represented as Mean {plus minus} SEM, unpaired two-tailed Student's t-test, ****p < 0.0001."

Figure EV3 legend - again, too much derail that is for Methods and already present in panels, μ g/ml puromycin concentration is for methods, whereas "for 0 min, 30 min or 60 min in HeLa WT and NAT10 KO cells" is reflected in the figure labels and not needed here (repeated 3 times within the same legend). Please check other legends as well.

Fra: e.schnapp@emboreports.org <e.schnapp@emboreports.org>

Sendt: 5. februar 2024 10:55

Til: Ulf Andersson Vang Ørom <ulf.orum@mbg.au.dk>

Emne: EMBOR: EMBOR-2023-57493V2 (N4-acetylcytidine (ac4C) promo...)

Dear Ulf,

We have now received the comments from referee 3 on your revised ms, I paste them below.

Please also address these comments in your final version of the ms. If you have any questions or comments regarding the final referee report, please let me know.

Referee 3:

While the revised submission from Kudrin et al presents some improvements from the previous version of the manuscript, there are still important flaws in how the data is presented and new questions raised by some of the data additions.

-In Figure 1, the data presentation is incomplete. For the imaging panels in 1A and 1B, there should be images shown for stressed and non-stressed conditions for each marker antibody tested. Figure 1 should also contain a quantification of the total ac4C signal that is colocalizing with G3BP-marked SGs in parental vs Nat10 KO cells.

-In supplemental Figure 1C, it seems that Nat10 KO cells are in fact not KO, since there is still a substantial Nat10 signal observed by Western blotting. How do the authors interpret this? This seems to put in doubt many of the conclusions of the paper in relation to Nat10 deficiency.

-The graphs presented in panels Figure 3D-G are hard to understand as explained in the figure legends, in particular with regards to the log₂FC axis. It should be clearly stated that the Cyto/SG ratio is being quantified.

-In Figure 4, the smFISH data show that all transcripts tested exhibit some extent of SG colocalization, both in parental and Nat10 KO cells. Indeed, if one looks at the quantifications provided by the authors in panels 4D and 4E, while the levels of mRNA recruitment to SGs seems lower in Nat10 KO specimens, there is still a clear induction of co-localization by longer exposure to arsenite (i.e. approximate 2-fold change between 30min and 60min exposure in both control and Nat10 KO), so ac4C is clearly not essential for stress-induced mRNA recruitment to SGs. Finally, the authors should also provide detailed quantification of localization properties of GAPDH mRNA, for some reason this was omitted from the figure.

-Figure 5 presents perhaps the most compelling data in the paper, revealing a loss of Nop58 localization to SGs in response to Nat10 KO. However, considering the data showing that Nat10 is still expressed in KO cells, this becomes difficult to interpret.

We would like to thank the editor and the reviewers for their comments, that we have incorporated in the revised manuscript as detailed below.

Editorial requests:

- Your ms has 5 main figures but is layed out as a full article. Please either combine the results and discussion sections or add one more main figure. You can find more info about our article types in our guide to authors online.

We have divided Figure 3 into two to make the results easier to understand and meet the criteria of 6 figures for a full article.

- Please add up to 5 keywords to the ms file.

We have added keywords to the manuscript.

- Please rename the conflict of interest subheading to "Disclosure Statement and Competing Interests"

Conflict of interest statement has been renamed.

- Please remove the author credits from the ms file. All credits need to be entered during ms submission now.

Author credit has been removed from the revised manuscript.

- Please correct the reference format to the EMBO reports style. It needs to be alphabetical and DOIs should only be used for preprints and datasets that have not been published yet.

Reference format has been changed to match that of EMBO reports.

- Figure 5 looks more like landscape than portrait format, it would be better if you could move panels F-H to below the other panels.

Figure 5 has been changed to a portrait format, it is now Figure 6.

- Fig S2A is called out but there is no such figure - please correct.

We have changed the call out to Fig EV2A.

- Tables EV1-EV3 are Datasets and need to be uploaded and renamed as such (Dataset EV1, etc); their legends should be removed from the ms file and provided in each Excel file as a separate sheet/tab; the callouts in the ms also need to be corrected.

We have changed the type of upload for EV1 to EV3 to datasets and renamed and formatted them as such.

- This new panel in the source data (SD) checklist needs to be corrected: S1F - are you referring to Figure EV1F?

The source data checklist has been corrected to Fig EV1F.

- Please rename Summary to Abstract

Summary has been renamed to abstract.

- The manuscript sections should be in the following order: Title page - Abstract & Keywords - Introduction - Results - Discussion - Materials & Methods - Data Availability - Acknowledgments - Disclosure Statement & Competing Interests - References - Figure Legends - Tables with legends - Expanded View Figure Legends.

We have updated the revised manuscript to follow this order of the sections.

- Please address the following comments by our data editors on the figure legends:

1. Please indicate the statistical test used for data analysis in the legends of figures 2a-b; 3h.
2. Please note that in figures 4e; EV 3e-f; there is a mismatch between the annotated p values in the figure legend and the annotated p values in the figure file that should be corrected.
3. Please note that the scale bar needs to be defined for figures EV 3c-d.

The comments from your data editors have been addressed in the revised manuscript.

I would like to suggest some minor changes to the abstract. Please let me know whether you agree with the following:

Stress granules are an integral part of the stress response that are formed from non-translating mRNAs aggregated with proteins. While much is known about stress granules, the factors that drive their mRNA localization are incompletely described. Modification of mRNA can alter the properties of the nucleobases and affect processes such as translation, splicing and localization of individual transcripts. Here, we show that the RNA modification N4-acetylcytidine (ac4C) on mRNA associates with transcripts enriched in stress granules and that stress granule localized transcripts with ac4C are specifically translationally regulated. We also show that ac4C on mRNA can mediate co-localization of the protein NOP58 to stress granules. Our results suggest that acetylation of mRNA regulates localization of both stress-sensitive transcripts and RNA-binding proteins to stress granules and adds to our understanding of the molecular mechanisms responsible for stress granule formation.

We agree with the changes suggested and have included the updated abstract in the revised manuscript.

Referee #1:

This revised manuscript from Kudrin and colleagues is substantially improved and all my major concerns have been adequately addressed. The authors clearly demonstrate the

enrichment of acetylated RNAs in SGs and the results represent a significant advance in the fields of RNA modifications and translational regulation. The association between ac4C and SG is intriguing and will certainly spark additional investigations. The revised manuscript is also improved in its general readability. However, I recommend a couple additional clarifications to make the manuscript more accessible to its readers, as outlined below. In addition, Table EV2 is insufficiently annotated. With these changes, I recommend this manuscript for publication.

Line 69-70. It is a stretch to propose that an additional RNA acetyltransferase exist based on the current state of the literature. Certainly, NAT10 depletion results in a profound decrease in ac4C levels at all tested RNA substrates. This is inconsistent with functionally redundant factors. Notably, Arango et al., 2018 showed that residual NAT10 is retained in CRISPR Cas9 deleted HeLa due to splicing around the targeted exon. It is possible that the low level of acetylation in the NAT10 deleted cells stems from the mis spliced protein. I recommend that the authors reword this section to also offer this alternative explanation.

We have rephrased to state that NAT10 is the only known ac4C acetyltransferase.

Fig. EV1e. I'm confused by the retention of ac4C signal in the nucleus after treatment with RNase. Is this because the nuclear RNA is inaccessible, or because the antibody is non-specific? I recommend that the authors offer a potential explanation.

We have added comments on this observation to the manuscript.

Table EV2. It is really not clear what is being presented in this table. What do the various columns mean? An additional tab with sample information should be provided. Were all the RNAs combined and used as bait? Was unmodified RNA used as a control? What control is used for non-specific binding? This table requires additional annotation prior to publication.

We have added more details to the descriptive tab in the Dataset EV2.

Referee #2:

I believe the authors addressed the vast majority of my comments however polishing of text is required, as stated in my original report, the manuscript can be made more concise and readability can be improved, some suggestions are below.

Fig1A: it is good to add RNAs but they can be barely read in purple, please revise colour scheme for this panel.

We have revised color scheme throughout the revised manuscript to make sure highlights are visible.

A bar with "NaAsO₂ treatment" label in it was added to many figures. However, this is confusing - this kind of bar would normally indicate an increased concentration, whereas the concentration is constant but the time increases.

We thank for this comment and have corrected the figure for easier interpretation.

Running title: too bold, this modification is not sufficient to localise mRNA to SGs, it has a modulatory effect.

We have modified the running title to attenuate the message.

Abstract: "In addition, we show that ac4C on mRNA can mediate co-localization of the protein NOP58 to stress granules." - localisation rather than co-localisation I believe.

We have changed to localization.

Lines 67-70 and 75-79 are essentially repetition. I suggest getting rid of the first mention and simply state that "In human, ac4C is deposited primarily by the acetyltransferase NAT10 (N-acetyltransferase 10) although alternative acetyltransferase(s) for RNA likely also exist." And keep lines 75-79.

We thank for this observation and have rephrased the manuscript not to repeat the same twice.

Manuscript still requires proofreading. E.g. line 83 "with both the SG markers G3BP and PABP" should be "for both SG markers G3BP and PABP" etc.

We have included this suggested change in the revised manuscript.

Line 89 - indispensable replace with essential

We have included essential.

G3BP or G3BP1 - needs consistency. Likewise, SG and SGs.

We have edited the manuscript for consistency and use G3BP and SG throughout.

Line 94 - "in upon arsenite stress." - remove 'in'

We removed in from the revised manuscript.

Section 2 and 3 'Isolation of SG associated transcripts from WT and NAT10 depleted cells' and 'The stress granule transcriptome is acetylated' should be merged and section 3 subheading kept.

We have merged the subsections as proposed by the reviewer.

Line 145 - 'transcripts'

Has been corrected.

Line 149 - I suggest deleting 'with less pronounced SG transcriptome', it sounds confusing.

The phrasing has been deleted as suggested.

Line 232: this sentence sounds awkward, please rephrase.

We have rephrased the sentence.

Line 268: NOP58 protein is binding to acetylated mRNA and is localized to SG - this section should be merged with the previous one, it does not require a separate subheading.

We have merged the sections as suggested.

Figure 3 legend is still too long and contains too much non-essential data description, please condense.

We have read through the legends and corrected and abbreviated where suitable.

Figure 4 legend - please revise to condense, e.g. "smFISH, coupled to G3BP-IF was used to visualise the core transcripts AHNAK and MKI67 and the control transcript GAPDH in SGs (a, b and c, respectively). Red is smFISH probe, green is G3BP and blue is DAPI. Scale bar, 10 μ m. Quantification of the fraction of AHNAK (d) and MKI67 (e) mRNAs in SGs per cytoplasm is also shown. 20 cells per each condition were counted (circles). Data is represented as Mean {plus minus} SEM, unpaired two-tailed Student's t-test, **** $p < 0.0001$."

We have read through the legends and corrected and abbreviated where suitable.

Figure EV3 legend - again, too much detail that is for Methods and already present in panels, μ g/ml puromycin concentration is for methods, whereas "for 0 min, 30 min or 60 min in HeLa WT and NAT10 KO cells" is reflected in the figure labels and not needed here (repeated 3 times within the same legend). Please check other legends as well.

We have read through the legends and corrected and abbreviated where suitable.

Referee #3

In Figure 1, the data presentation is incomplete. For the imaging panels in 1A and 1B, there should be images shown for stressed and non-stressed conditions for each marker antibody tested. Figure 1 should also contain a quantification of the total ac4C signal that is colocalizing with G3BP-marked SGs in parental vs Nat10 KO cells.

In the revised version, we have included all marker antibodies for both stressed and unstressed conditions in Figure 1 as well as EV1. Regarding total ac4C signal colocalizing with SG, we had initially shown images of entire cells and staining of ac4C in both control and

stressed cells, WT and NAT10, side by side. Now they are in Figure 1B and Figure EV1D. Here we see a clear reduction of ac4C in the entire cell, including SG.

We see the reduction of ac4C in whole cells by comparing mass spec for WT and NAT10 KO in Figure EV1B. To Elaborate on this, we purified SG from WT cells to show an enrichment of ac4C over other RNA modifications in Figure 1D and Figure EV1F-J, to make sure it is not an artefact of disrupted SG formation.

In supplemental Figure 1C, it seems that Nat10 KO cells are in fact not KO, since there is still a substantial Nat10 signal observed by Western blotting. How do the authors interpret this? This seems to put in doubt many of the conclusions of the paper in relation to Nat10 deficiency.

We have discussed this in the manuscript, where we state how the cells are generated, *i.e.* not a KO of the entire NAT10 but introduction of a mis-sense deletion making the enzyme non-functional.

The graphs presented in panels Figure 3D-G are hard to understand as explained in the figure legends, in particular with regards to the log2FC axis. It should be clearly stated that the Cyto/SG ratio is being quantified.

We have improved the legends to emphasize that the log2FC axes show Cyto/SG ratio.

In Figure4, the smFISH data show that all transcripts tested exhibit some extent of SG colocalization, both in parental and Nat10 KO cells. Indeed, if one looks at the quantifications provided by the authors in panels 4D and 4E, while the levels of mRNA recruitment to SGs seems lower in Nat10 KO specimens, there is still a clear induction of co-localization by longer exposure to arsenite (*i.e.* approximate 2-fold change between 30min and 60min exposure in both control and Nat10 KO), so ac4C is clearly not essential for stress-induced mRNA recruitment to SGs. Finally, the authors should also provide detailed quantification of localization properties of GAPDH mRNA, for some reason this was omitted from the figure.

We have included in the revised manuscript that ac4C promotes SG localization, but is not essential for SG formation. We did not include GAPDH quantification as there is hardly any enrichment in SG compared to the levels in cytoplasm and the numbers would almost disappear in comparison to AHNAK and MKI67.

Figure 5 presents perhaps the most compelling data in the paper, revealing a loss of Nop58 localization to SGs in response to Nat10 KO. However, considering the data showing that Nat10 is still expressed in KO cells, this becomes difficult to interpret.

As for point 2, NAT10 is expressed as a non-functional enzyme as also demonstrated by the high impact on ac4C levels. In the revised manuscript we state that the difference in NOP58 localization is due to the loss of ac4C, the levels of which we clearly demonstrate.

Dr. Ulf Orom
Department of Molecular Biology and Genetics
Universitetsbyen 81
Aarhus, Aarhus 8000
Denmark

Dear Ulf,

I am very pleased to accept your manuscript for publication in the next available issue of EMBO reports. Thank you for your contribution to our journal.
